# Cell biological analysis reveals an essential role for *Pfcerli2* in erythrocyte invasion by malaria parasites

Benjamin Liffner [1,13], Juan Miguel Balbin[1,13], Gerald J. Shami[2], Ghizal Siddiqui [3], Jan Strauss [4,5,6,11], Sonja Frölich [1], Gary K. Heinemann [7], Ella May Edwards[1], Arne Alder [4,5,6], Jan Stephan Wichers [4,5], Darren J. Creek[3], Leann Tilley [2], Matthew W. A. Dixon [2,8,12], Tim-Wolf Gilberger[4,5,6] & Danny W. Wilson [1,9,10 ✉]

Merozoite invasion of host red blood cells (RBCs) is essential for survival of the human malaria parasite *Plasmodium falciparum*. Proteins involved with RBC binding and invasion are secreted from dual-club shaped organelles at the apical tip of the merozoite called the rhoptries. Here we characterise *P. falciparum* Cytosolically Exposed Rhoptry Leaflet Interacting protein 2 (PfCERLI2), as a rhoptry bulb protein that is essential for merozoite invasion. Phylogenetic analyses show that *cerli2* arose through an ancestral gene duplication of *cerli1*. We show that PfCERLI2 is essential for blood-stage growth and localises to the cytosolic face of the rhoptry bulb. Inducible knockdown of PfCERLI2 led to a proportion of merozoites failing to invade and was associated with elongation of the rhoptry organelle during merozoite development and inhibition of rhoptry antigen processing. These findings identify PfCERLI2 as a protein that has key roles in rhoptry biology during merozoite invasion.

[1] Research Centre for Infectious Diseases, School of Biological Sciences, University of Adelaide, Adelaide 5005, Australia. [2] Department of Biochemistry and Pharmacology, Bio21 Molecular Science and Biotechnology Institute, The University of Melbourne, Melbourne, VIC 3010, Australia. [3] Dept Drug Delivery, Disposition and Dynamics, Monash Institute of Pharmaceutical Sciences, Monash University, Parkville, VIC 3052, Australia. [4] Centre for Structural Systems Biology, 22607 Hamburg, Germany. [5] Bernhard Nocht Institute for Tropical Medicine, 20359 Hamburg, Germany. [6] Biology Department, University of Hamburg, 20146 Hamburg, Germany. [7] Experimental Therapeutics Laboratory, Clinical and Health Science Unit, University of South Australia, Adelaide, SA 5000, Australia. [8] Department of Infectious Diseases, Peter Doherty Institute for Infection and Immunity, University of Melbourne, Melbourne, 3000, Australia. [9] Institute for Photonics and Advanced Sensing, University of Adelaide, Adelaide 5005, Australia. [10] Burnet Institute, 85 Commercial Road, Melbourne, VIC 3004, Australia. [11] Present address: GEOMAR Helmholtz Centre for Ocean Research Kiel, 24105 Kiel, Germany. [12] Present address: Division of Infectious Diseases and Immune Defence, Walter and Eliza Hall Institute of Medical Research, Parkville, 3052, Australia. [13] These authors contributed equally: Benjamin Liffner, Juan Miguel Balbin. ✉email: danny.wilson@adelaide.edu.au

*P*lasmodium falciparum, a human malaria parasite, is the cause of ~400,000 deaths each year; predominantly in children under the age of five[1]. *P. falciparum* is transmitted from *Anopheles* mosquitoes, its definitive host, to humans through the blood meal of an infected mosquito where invasive sporozoites are injected subcutaneously and migrate to the liver. Following invasion of liver hepatocytes, the parasite develops into thousands of daughter merozoites inside a hepatic schizont, which ruptures and releases the daughter merozoites into the blood stream where they invade red blood cells (RBCs)[2]. Over the following ~48 h, the parasite develops inside the RBC until it forms 16–32 new daughter merozoites inside of a schizont that ruptures, releasing the merozoites that go on to infect new RBCs[2]. Replication of *P. falciparum* in this asexual blood-stage of the lifecycle is the cause of all the clinical symptoms of malaria.

RBC invasion by merozoites occurs over a period of ~30 s and involves the attachment of the merozoite to the RBC, which then reorients so its apical tip contacts the RBC membrane[3]. Specialised invasion organelles, known as the rhoptries and micronemes, then secrete their contents to form an irreversible tight junction before the merozoite pulls the RBC plasma membrane around itself, forming a parasitophorous vacuole (PV) and completing invasion[4]. Each merozoite has two rhoptries, the largest of the invasion organelles, which are club-shaped and divided into a bulb and neck, with the neck positioned at the apical tip of the merozoite[5]. Rhoptry neck proteins are released early in the invasion process and mediate initial attachment to the RBC and formation of the tight junction[6]. Rhoptry bulb contents are secreted following junction formation and are typically involved in establishing the PV[7].

Most rhoptry luminal proteins exclusively localise to the rhoptry bulb or neck, with minimal overlap between the two[8,9]. During the process of RBC invasion, the neck of the rhoptries fuse to the parasite plasma membrane (PPM) to allow secretion of rhoptry contents[9]. In addition, the dual-club-shaped rhoptries fuse from neck to bulb, leaving a singular rhoptry that retains its neck and bulb structure[5,9]. What controls both PPM and rhoptry fusion has not been elucidated.

It is likely that rhoptry and PPM fusion are controlled by proteins that localise to the cytosolic face of the rhoptry membrane, as they can contact both membranes to facilitate their fusion. To date, however, only four proteins in *P. falciparum* have been shown to localise to the cytosolic face of the rhoptry membrane[9]: Armadillo repeats only (ARO)[10], ARO interacting protein (AIP)[11], Adenylate cyclase beta (ACß)[12], and Cytosolically exposed rhoptry leaflet interacting protein 1 (CERLI1; also known as Rhoptry apical surface protein (RASP) 2)[13,14]. Here we describe PfCERLI2 (Pf3D7_0405200), a paralogue of *Pfcerli1*, which localises to the cytosolic face of the rhoptry bulb membrane and is essential for merozoite invasion of RBCs.

## Results

**PfCERLI2 is conserved among Apicomplexa and arose from an ancestral gene duplication.** *Plasmodium falciparum* Cytosolically Exposed Rhoptry Leaflet Interacting protein 2 (PfCERLI2) (Pf3D7_0405200) is a protein of 579 amino acids in *P. falciparum* isolate 3D7. PfCERLI2 contains a predicted lipid-calcium binding C2 domain towards its N-terminus and possesses a decapeptide tandem repeat (consensus sequence QTEIkNDhi; upper case = fixed amino acid, lower case = variable amino acid) at its C-terminus, with the number of repeats ranging from 10 to 20 between different *P. falciparum* isolates (Fig. 1a; Supplementary Data 1). PfCERLI2 shares homology (~20% amino acid identity) (Supplementary Fig. 1) with the recently characterised protein PfCERLI1[13,14], with PfCERLI2 being the most similar protein to

PfCERLI1 in the *P. falciparum* 3D7 proteome on PlasmoDB[15]. Notably, PfCERLI1 and PfCERLI2 both possess a C2 domain towards their N-terminus as well as a short but highly conserved motif we have termed PHIS, which has the consensus sequence PHIS[-]xxP ([-] = negatively charged) in *P. falciparum*. PfCERLI2 is highly conserved amongst *Laverania*, a subgenus that comprises *P. falciparum* and its closest ape-infecting relatives (>90% amino acid identity with *Laverania* homologues), and shares ~65% amino acid identity amongst more distantly related *Plasmodium* spp. (Fig. 1b; Supplementary Fig. 2). Notably, PfCERLI2 shares high similarity with its homologues over the first ~420 amino acids of its sequence, but this similarity decreases significantly at the *Laverania*-specific C-terminal repeat region.

All organisms that diverged from *Plasmodium* more recently than *Cryptosporidium* contain homologues of both *Pfcerli1* and *Pfcerli2*, while earlier diverging organisms, such as *Gregarina niphandrodes* and the Chromerid *Vitrella brassicaformis*, contain only a single identifiable homologue (Fig. 1c; Supplementary Fig. 3); suggesting that *Pfcerli1* and *Pfcerli2* are paralogues that arose from an ancestral gene duplication event. As genomic DNA (14.6%) and amino acid (22.9%) sequence identities were low across Apicomplexa and Chromera, analysis of sequence conservation, outside of *Plasmodium*, was not able to identify the direct orthologues of either *Pfcerli1* or *Pfcerli2* outside the *Plasmodium* genus (Supplementary Fig. 4a). Therefore, we compared the gene structure of *Pfcerli1* and *Pfcerli2*, along with homologous sequences in Apicomplexa and Chromerids (Supplementary Data 2) to identify the putative *cerli1* and *cerli2* orthologues across distantly related species. The comparison of *cerli1* and *cerli2* sequences across *Plasmodium* spp. indicated that *cerli1* orthologues have fewer introns and decreased exon length than the orthologues of *cerli2* (Supplementary Data 3). This gene structure pattern was also observed for homologues outside the *Plasmodium* genus, allowing for a classification of more divergent homologues as the direct orthologue of either *Pfcerli1* or *Pfcerli2*. Using this approach, we determined that the direct orthologue of *Pfcerli2* in *Toxoplasma gondii* is TGME49_315160 (TgRASP2), which was previously reported to be the direct orthologue of *Pfcerli1*[14]. In addition, our classification suggests that the direct orthologue of *Pfcerli1* is TGME49_235130 (TgRASP1). As Apicomplexa and Chromerids that are distantly related to *P. falciparum* only possess a single homologue of *Pfcerli1* and *Pfcerli2*, we compared their gene structure to determine which gene is more similar to the conserved ancestral state and which one represents a more diverged duplicated copy. All *Cryptosporidium* spp. homologues have an intron number and exon length similar to the *cerli1* lineage, suggesting that *cerli1* represents the ancestral locus and *cerli2* the duplicated locus. There is no consistent trend in gene structure in more distantly related organisms (Gregarines or Chromerids), suggesting the gene structure may not have been conserved in more divergent organisms.

While the overall sequence identity between PfCERLI1, PfCERLI2, and their homologues is low, all contain the highly conserved PHIS motif with the consensus sequence PHPSECxP (when comparing all apicomplexan/chromerid homologues; Supplementary Fig. 4b) or PHIS[-]xxP (when comparing *P. falciparum* sequences; Supplementary Fig. 4c). Notably, the first proline is fully conserved across all analysed sequences, and the histidine residue is mutated only in *Gregarina niphandrodes*. A second proline lies at position 8 in all CERLI homologues in *Plasmodium* spp., while in all other apicomplexans outside *Plasmodium*, the second proline lies at position 3. Such high sequence conservation across highly divergent organisms suggests this motif is likely to be critical for the physiological function of

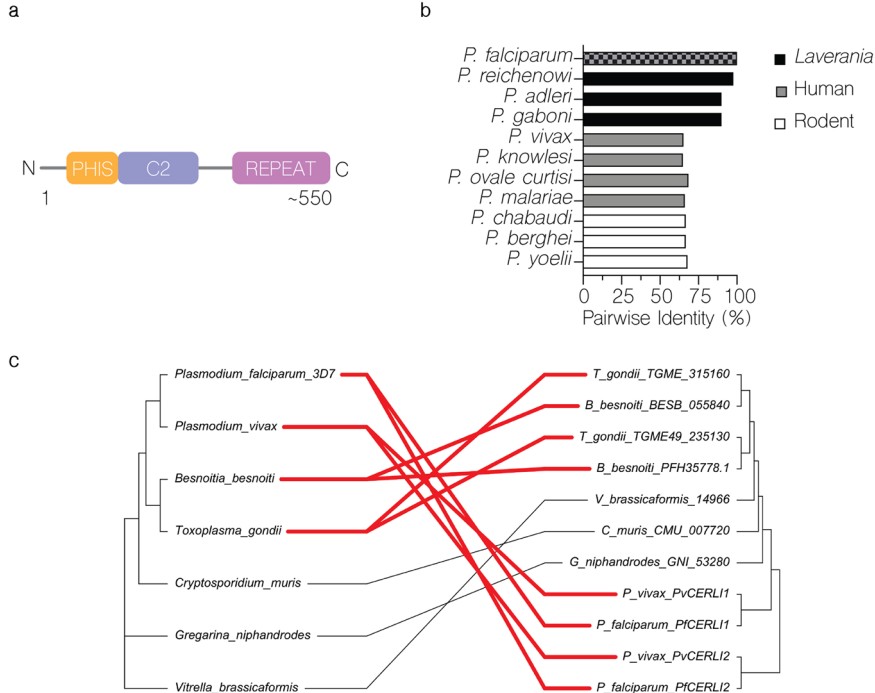

**Fig. 1 PfCERLI2 is conserved among Apicomplexa and may have evolved from an ancestral gene duplication. a** PfCERLI2 is a protein of 579 amino acids in *P. falciparum* 3D7. Towards its N-terminus PfCERLI2 contains a motif with the consensus sequence PHIS[-]xxP we have termed PHIS, a C2 domain, and a decapeptide tandem repeat with the consensus sequence QTEIkNDhi at its C-terminus. Repeat number (10–20), and therefore PfCERLI2 amino acid length (559–659), is highly variable between *P. falciparum* isolates. **b** Amino acid sequence identity for *Plasmodium* spp. PfCERLI2 orthologues in Laverania, human-infecting, and rodent-infecting parasites was compared using multiple pairwise alignments. **c** Tanglegram comparing general evolutionary relationships between selected Apicomplexa and Chromerids, as described in ref. [71] (left), with phylogenetic tree constructed with PfCERLI1, PfCERLI2 and homologus sequences retrieved from EuPathDB using the unweighted pair group method with arithmetic mean (UPGMA) method (right). Branch length of UPGAM tree corresponds to amino acid substitutions per site. Taxa containing CERLI1 and CERLI2 homologues are joined by red edges, while taxa with a single CERLI are joined by black edges to visualise timing of ancestral gene duplication giving rise to CERLI2.

PfCERLI1, PfCERLI2 and their homologues. Similar amino acid motifs exist in other *P. falciparum* proteins; including the ring-expressed surface antigens (RESA, RESA2, RESA3 and RESA-like), iron superoxide dismutase (FeSoD) and ATP-dependent Clp protease proteolytic subunit (ClpR) (Supplementary Data 4; Supplementary Fig. 4c). Although the PHIS motif is a signature of PfCERLI1 and PfCERLI2, and their apicomplexan homologues, nothing is currently known about the function of this motif or whether it has a related role in the other *P. falciparum* PHIS-containing proteins.

With our analysis supporting that *Pfcerli2* arose from a gene duplication event, we next assessed the selection pressure acting on the *Pfcerli1* and *Pfcerli2* sequences. To investigate this, we determined Ka/Ks ratios, which calculate the number of non-synonymous amino acid substitutions (Ka) relative to the number of synonymous substitutions (Ks). *Pfcerli1* and *Pfcerli2* across *Plasmodium* spp. had Ka/Ks of 0.091 and 0.29, respectively, indicating strong negative selection (favouring low sequence diversity), which is consistent with their essentiality for *P. falciparum*[13,14,16] and *P. berghei*[17] blood-stage survival. This negative selection in both lineages may be indicative of subfunctionalisation in Apicomplexa, where both paralogues have maintained a subset of the function of the ancestral gene.

Interestingly, homologues of PfCERLI2 within the *Plasmodium* genus no longer contain a Pleckstrin homology (PH) domain that is predicted for homologous sequences outside the *Plasmodium* genus (Supplementary Fig. 5) and has been shown to be involved in rhoptry localisation and secretion for TgRASP2[14]. Analysis of predicted protein structures for PfCERLI1 and PfCERLI2 homologues outside the *Plasmodium* genus show that this PH

domain is typically predicted to contain two sets of anti-parallel beta-sheets separated by a short alpha-helix, followed by a C-terminal helix (Supplementary Fig. 6). By contrast, *Plasmodium* homologues of CERLI2 are predicted to have two beta-sheets in the region corresponding to the PH domain in CERLI2 of other species and PfCERLI1, but this region also has an expanded helix and disordered region between the beta-sheets (Supplementary Fig. 6). In addition, PfCERLI2 is predicted to contain a shorter C-terminal helix (Supplementary Fig. 6). The expansion between the beta-sheets is predicted to disrupt the interaction between them that is required for canonical PH domain formation. Therefore, although elements of the PH domain are predicted to remain (beta-sheets) structural predictions do not annotate a PH domain structure in this region and it is questionable whether this domain structure is functional in PfCERLI2.

**PfCERLI2 contains a variable copy number decapeptide tandem repeat.** All PfCERLI2 homologues in *Laverania* contain a C-terminal decapeptide tandem repeat (Supplementary Figs. 7 and 8), with the consensus sequence QTIEIkNDhi in *P. falciparum*, which is not present in any homologues outside the subgenus (Supplementary Data 1)[18]. The repeat number is variable both between *P. falciparum* isolates (e.g. 10 repeats in isolate 7G8 and 20 repeats in isolate Dd2), and between *Laverania* (9 repeats in *P. praefalciparum* and 4 repeats in *P. adleri*). It is not clear what the function of this repeat is, or what drives changes in its number. However, we noticed that *P. falciparum* has a higher number of repeats than other *Laverania*, with organisms more

closely related to *P. falciparum* also displaying higher repeat numbers than those more distantly related (Supplementary Fig. 7; Supplementary Data 1). The full sequence of *Pfcerli2* is under strong negative selection across *Laverania*, with a Ka/Ks ratio of 0.29. By contrast, the repeat region of *Pfcerli2* when comparing across *Laverania* has a Ka/Ks ratio of 1.43 indicating that positive selection (favouring sequence diversity) is occurring in the decapeptide repeat region of PfCERLI2 and Laveranian homologues.

**PfCERLI2 is essential for blood-stage growth.** PfCERLI2 has previously been suggested to be essential by a saturation mutagenesis screen[16], and its homologue in *P. berghei* has also been shown to be essential for blood-stage growth[17]. To confirm this in *P. falciparum*, we targeted *Pfcerli2* by selection-linked integration targeted gene disruption (SLI-TGD)[19]. Multiple attempts to knockout *Pfcerli2* failed confirming that *Pfcerli2* has an important role in blood-stage growth (Fig. 2a). Next, we placed *Pfcerli2* under control of the glucosamine (GLCN) inducible *GlmS* ribozyme[20], whereby addition of GLCN leads to specific degradation of *Pfcerli2* mRNA (Fig. 2b, c). In addition, we introduced a haemagglutinin (HA) tag onto the C-terminus of PfCERLI2, allowing detection using anti-HA antibodies. Using these PfCERLI2[HAGlmS] parasites, we harvested parasites at either ring, trophozoite or schizont stages and probed for PfCERLI2 by western blot (Fig. 2d). Concordant with previously published transcriptomic data[21,22], PfCERLI2 was most highly expressed in schizonts. When PfCERLI2[HAGlmS] parasites were treated from ring stages to schizonts with 2.5 mM GLCN, the expression of PfCERLI2 was reduced by 72% (±3.8% SEM) relative to untreated controls (Fig. 2e), showing that PfCERLI2 protein expression is tunable using the *GlmS* system.

To determine whether knockdown of PfCERLI2 altered parasite growth, we treated ring-stage PfCERLI2[HAGlmS] and 3D7 WT parasites with increasing concentrations of GLCN and measured trophozoite-stage parasitaemia by flow cytometry 72 h later (Fig. 2f). At 5 mM GLCN there was ~77% (±5.1% SEM) decrease in the growth of PfCERLI2[HAGlmS] parasites. However, at this high concentration of GLCN growth of 3D7 WT parasites was also affected. At a concentration of 2.5 mM GLCN, growth of PfCERLI2[HAGlmS] parasites was inhibited by ~53% (±6.0% SEM), with 3D7 WT parasites showing negligible off-target growth defects.

**PfCERLI2 knockdown inhibits merozoite invasion.** As PfCERLI2 was most highly expressed at schizont stages, and its knockdown inhibited growth, we hypothesised that PfCERLI2 could be involved in schizont development, rupture, or merozoite invasion. To quantify invasion, we transfected PfCERLI2[HAGlmS] parasites with a plasmid that expresses cytosolic GFP (PfCERLI2[HAGlmS/GFP]), which allows reliable detection of newly invaded ring stages by flow cytometry. Treating PfCERLI2[HAGlmS/GFP] parasites with increasing concentrations of GLCN from trophozoite stages for 24 h until early ring-stages the following cycle, we saw that PfCERLI2 knockdown inhibited invasion in a dose-dependent manner (Fig. 3a). The extent of invasion inhibition caused by PfCERLI2 knockdown was lower than growth inhibition over 72 h, however, it is likely that this difference is caused by the shorter GLCN treatment time used for assessing inhibition of invasion.

To determine if this invasion inhibition was specifically due to a defect in merozoite development, PfCERLI2[HAGlmS] schizonts either GLCN treated or untreated, were matured in the presence of the schizont egress inhibitor E64. Knockdown of PfCERLI2 did not result in a change in the morphological development of trophozoites (Supplementary Fig. 9a and b), nor in the number of

fully formed merozoites per schizont (Fig. 3b). The merozoites within the schizonts appeared morphologically normal (Fig. 3c). To determine whether PfCERLI2 knockdown altered the rate of schizont rupture, the ability of both GLCN treated and untreated PfCERLI2[HAGlmS] synchronised schizonts to rupture over a period of 6 h was quantified. PfCERLI2 knockdown had no influence on the rate of schizont rupture (Fig. 3d), suggesting knockdown has no influence on schizont development or egress. To test if PfCERLI2 knockdown directly inhibited the invasion of merozoites, we measured the number of free merozoites in the culture supernatant following invasion in both GLCN treated and untreated PfCERLI2[HAGlmS/GFP] parasites. PfCERLI2 knockdown led to an increase in the number of free merozoites (Fig. 3e), suggesting that a direct defect in the ability of merozoites to invade is the cause of PfCERLI2 knockdown-mediated growth inhibition.

The flow cytometry-based assay does not allow discrimination between free merozoites inhibited prior to or after (bound merozoites) tight junction formation. Therefore, we analysed Giemsa-stained thin smears of GLCN treated and untreated PfCERLI2[HAGlmS] ring-stage parasites and quantified the number of bound merozoites following schizont rupture (Fig. 3 f, g). PfCERLI2 knockdown led to a 49% (±12% SEM) increase in the percentage of merozoites bound to RBCs, suggesting that knockdown inhibits invasion following tight junction formation. Newly invaded ring-stage parasites (Fig. 3h) were also quantified, with GLCN treatment leading to a 41% (±5.9% SEM) reduction in newly invaded ring stages.

Although Giemsa smears demonstrated an increase in attached merozoites that had failed to invade with PfCERLI2 knockdown relative to control, they cannot accurately show whether the tight junction has formed. Therefore, we used a cytochalasin D-based strategy in fixed cells and analysed invading merozoites by Airyscan super-resolution microscopy (Fig. 3i–k). Viable GLCN treated and untreated PfCERLI2[HAGlmS] merozoites were purified and permitted to invade RBCs stained with CellTrace™ Far Red (Fig. 3i), and treated with or without cytochalasin D, a potent inhibitor of actin polymerisation which prevents completion of invasion[23]. CellTrace has been reported to not affect invasion and is a suitable stain to visualise the RBC[24]. Invasion scoring was performed and demonstrated that a large relative proportion of GLCN treated parasites were unable to fully invade the RBC (Fig. 3j), consistent with earlier results. Following from our Giemsa-based observations, GLCN merozoites appeared to be able to successfully reorientate their apical tip to the RBC surface and thus we speculate that PfCERLI2 knockdown may inhibit invasion downstream of this step, for example at tight junction formation. To quantify this observation, the diameter of AMA1 staining was measured and used as a proxy to determine progression through the tight junction ring—where a larger ring indicates successful progression. However, as the tight junction ring ratchets itself behind the merozoite at the completion of invasion, potentially becoming smaller, we have excluded data points from merozoites with an invasion score = 3 (fully invaded) from all samples for this analysis (full data in Supplementary Fig. 9c). Using this analysis, we found that PfCERLI2 knockdown parasites have a smaller AMA1 diameter compared to untreated parasites, results that mimic those of cytochalasin D treated parasites which blocks entry after formation of the tight junction (Fig. 3k). Put together, these data indicate that GLCN inducible PfCERLI2 knockdown inhibits invasion at, or prior to, the formation of the tight junction.

**PfCERLI2 localises to the rhoptry bulb.** The homology to PfCERLI1 as well as its functional implications in RBC invasion

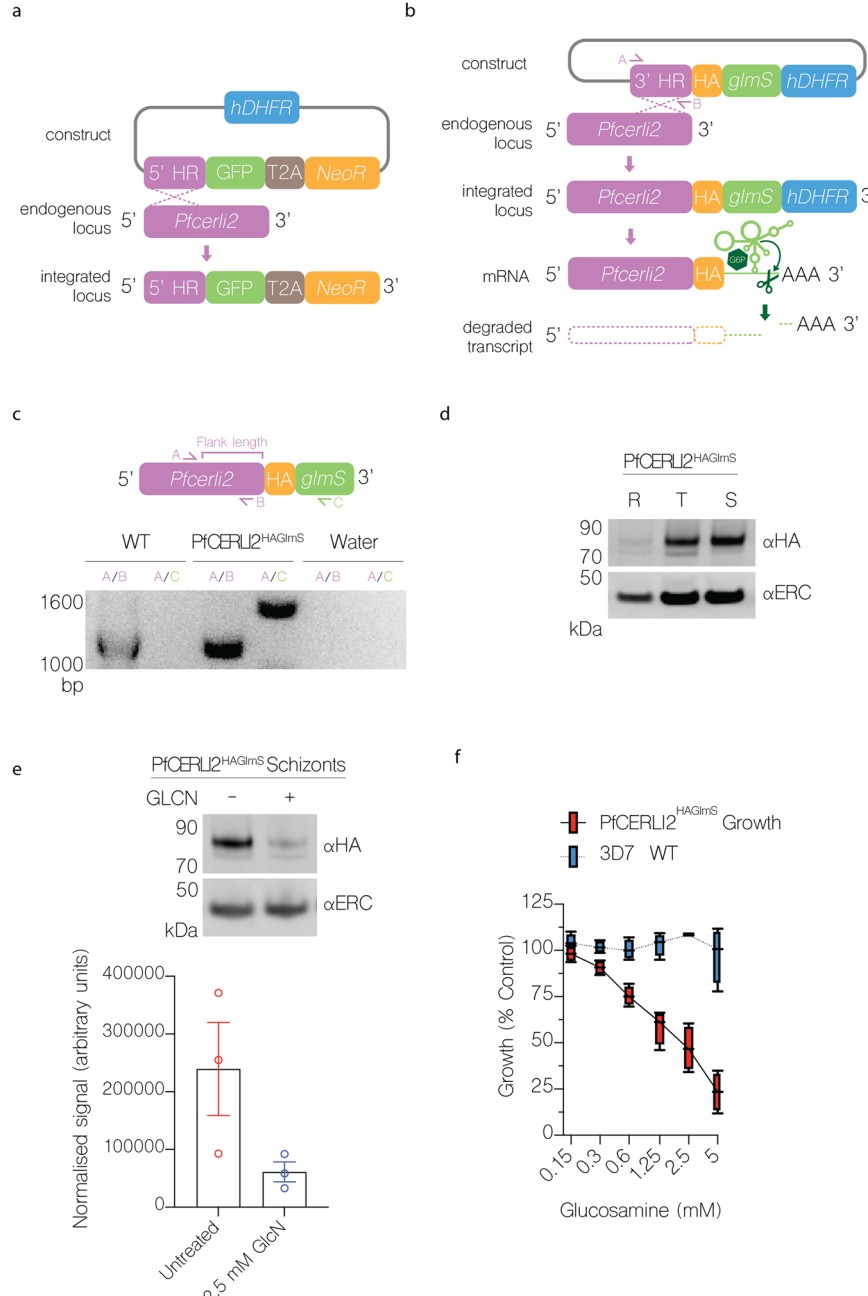

**Fig. 2 PfCERLI2 is essential for blood-stage growth. a** Schematic representation of the selection linked integration targeted gene disruption (SLI-TGD) system used for the attempted gene knockout of *Pfcerli2*. **b** Schematic representation of the haemagglutinin (HA) tag, and GlmS ribozyme system used to identify and knockdown PfCERLI2. A plasmid that contained a 3′ *Pfcerli2* flank (2154bp-2912bp) with an HA-tag and GlmS ribyzome was transfected into 3D7 parasites by single crossover recombination. Following mRNA production, glucosamine-6-phosphate (G6P) binds to the GlmS ribozyme, leading to the cleavage of *Pfcerli2* mRNA and subsequent protein knockdown. **c** To confirm integration of the *Pfcerli2*[HAGlmS] plasmid, gDNA was harvested from transfected parasites. A *Pfcerli2* specific forward primer, A, with either a *Pfcerli2* specific, B (WT), or GlmS specific, C (integrated), reverse primer was used to confirm WT DNA sequence or integration of the *Pfcerli2*[HAGlmS] plasmid. **d** Western blot of ring, trophozoite, or schizont stage PfCERLI2[HAGlmS] lysates probed with either anti-HA (PfCERLI2) or anti-ERC (loading control) antibodies. $N = 3$ biological replicates. **e** Western blot of PfCERLI2[HAGlmS] schizont lysates either GLCN treated (+) or untreated (−) and probed with anti-HA (PfCERLI2) normalised to ERC (loading control) antibodies. A graph quantifying the relative level of expression between treatments. $N = 3$ biological replicates, error bars = SEM. **f** PfCERLI2[HAGlmS] or 3D7 WT parasites were treated with increasing concentrations of GLCN for 72 h, with the trophozoite-stage parasitaemia determined by flow cytometry. Growth is expressed as a percentage of an untreated media control. $n = 4$, box represents the 25th to 75th percentile, median and maximum value are shown. X-axis presented as a log 2 scale for viewing purposes.

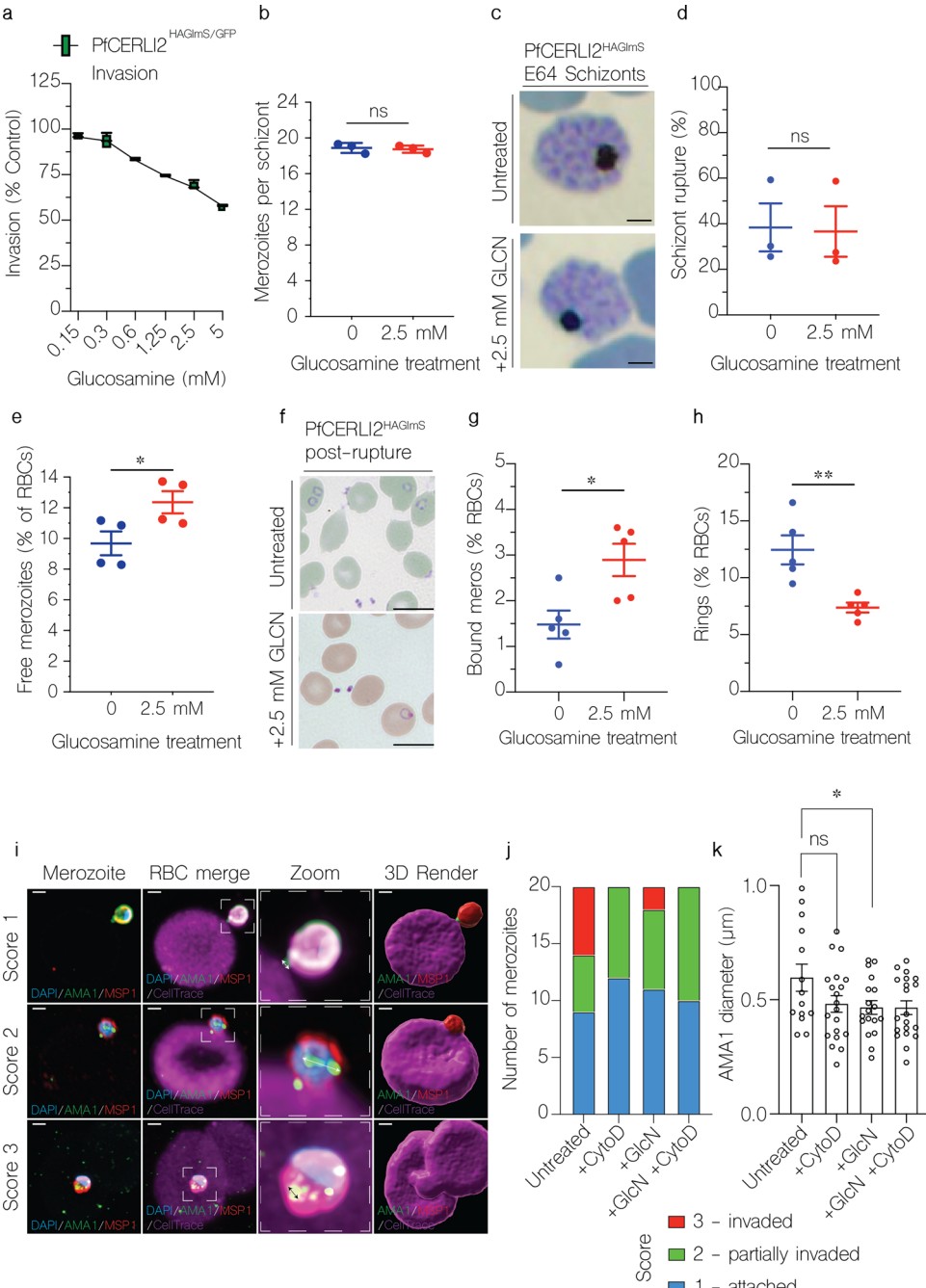

**Fig. 3 PfCERLI2 knockdown inhibits merozoite invasion. a** GFP-expressing PfCERLI2[HAGlmS/GFP] parasites were treated with a range of concentrations of GLCN from early trophozoite stages until early rings the following cycle (~24 h), with invasion assessed by flow cytometry as the number of newly invaded rings expressed as a percentage of an untreated media control. $n = 4$, box represents the 25th to 75th percentile, median and maximum value are shown. X-axis presented as log 2 scale for viewing purposes. **b** PfCERLI2[HAGlmS] parasites were either treated with GLCN from ring-stage to schizont stage or left untreated. Early schizonts were treated with the egress inhibitor E64 for ~5 h. Following treatment, cultures were smeared and the number of merozoites per schizont was determined by blinded microscopy analysis of Giemsa-stained blood smears. Each data-point represents the mean number of merozoites per schizont from 20 schizonts. $n = 3$, error bars = SEM, ns = $p > 0.05$ by unpaired $t$-test. **c** Percentage of schizonts that ruptured over a 6-h window in either the presence or absence of GLCN, as determined by flow cytometry. $n = 3$, error bars = SEM, ns = $p > 0.05$ by unpaired $t$-test. **d** During the invasion assay described in (**a**), the number of free merozoites was also quantified by flow cytometry, with results presented as % of total RBCs. $n = 4$, error bars = SEM, *$p < 0.05$ by unpaired $t$-test. **e** PfCERLI2[HAGlmS] schizonts, either in the presence or absence of GLCN, were matured in the presence of the schizont rupture inhibitor E64 before being fixed, Giemsa-stained and imaged by light microscopy. Scale bar = 2 µm. PfCERLI2[HAGlmS] parasites were treated with 2.5 mM GLCN or left untreated, with Giemsa smears made following schizont rupture (**f**) and the number of merozoites bound to RBCs (**g**) and newly invaded rings (**h**) quantified, Scale bar = 5 µm, $n = 5$, error bars = SEM, *$p < 0.05$, **$p < 0.01$ by unpaired $t$-test. **i** Representative IFAs of merozoites, across three defined invasion stages scored 1–3, stained with DAPI, anti-AMA1 and anti-MSP1-19 antibodies invading CellTrace-stained RBCs. Scale bar = 1 µm. **j** Number of merozoites blind-assessed for each invasion score. $n = 20$ merozoites in a single biological replicate. **k** AMA1 signal diameter was measured under blinded assessment—measurements were taken as shown on the double arrows in (**i**). $N = 20$ merozoites in a single biological replicate, error bars = SEM, *$p < 0.05$, ns = $p > 0.05$ by unpaired $t$-test.

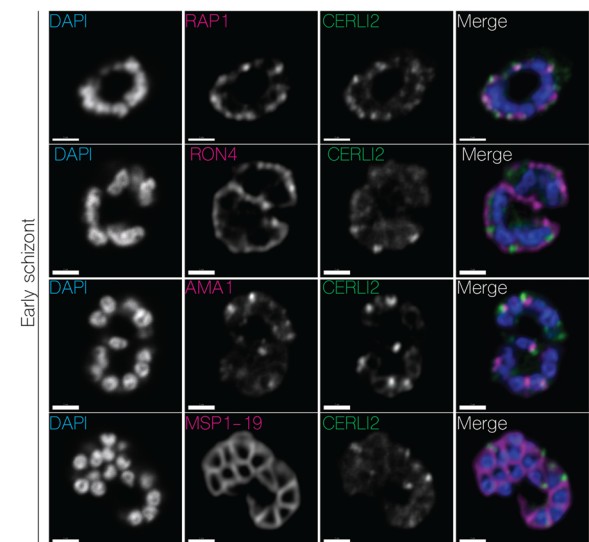

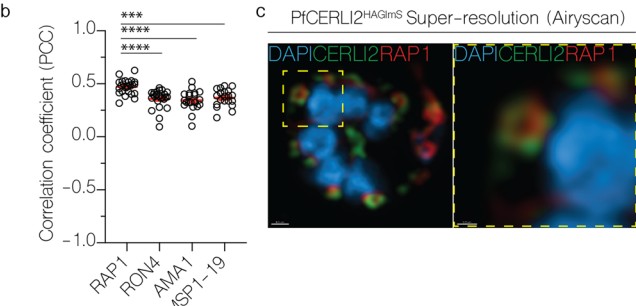

**Fig. 4 PfCERLI2 localises to the rhoptry bulb. a** 2D Confocal immunofluorescence microscopy of early PfCERLI2HAGlmS schizonts stained with DAPI (nucleus) and anti-HA (PfCERLI2) antibodies, along with antibodies to either RAP1 (rhoptry bulb), RON4 (rhoptry neck), AMA1 (micronemes), or MSP1-19 (merozoite surface). Scale bar = 2 µm. **b** Quantification of colocalisation between PfCERLI2 and the merozoite organelle markers RAP1, RON4, AMA1 and MSP1-19. Colocalisation quantified as Pearson's correlation coefficient (PCC) when the PfCERLI2 signal was defined as the region of interest. $n = 3$ biological replicates, 6 schizonts per replicate. Error bars = SEM, ***$p < 0.001$, ****$p < 0.0001$, by unpaired $t$-test. **c** Representative maximum-intensity projection of PfCERLI2HAGlmS parasites stained with antibodies to RAP1 (rhoptry bulb) and HA (CERLI2) and imaged using the super-resolution microscopy platform Airyscan. Yellow box indicates the zoom area for the right-hand panel.

suggested PfCERLI2 may have a rhoptry localisation. In our numerous attempts to localise PfCERLI2, we experienced that conventional antibody staining methods could not reliably visualise any foci directed to its HA-tag outside of background signal. After failing to obtain a consistent PfCERLI2 localisation, likely due to the low peak expression level of *Pfcerli2*[21], we investigated the cellular distribution of PfCERLI2 using a high-affinity anti-HA biotin system to significantly bolster the signal[25]. Despite its peak transcription late in schizont development, we observed that PfCERLI2 foci were brightest in early schizonts (~44 h post invasion) (Fig. 4a), but largely absent in fully mature E64-treated schizonts (~48 h post invasion) (Supplementary Fig. 10). Harvesting early schizonts (~44 h post invasion), we analysed subcellular organisation of CERLI2 by comparing it to known markers of the rhoptry bulb (RAP1), rhoptry neck (RON4), micronemes (AMA1), and merozoite surface (MSP1-19)

(Fig. 4a and b). In these parasites, PfCERLI2 showed the strongest colocalisation with the rhoptry bulb marker RAP1, suggesting PfCERLI2 likely localises to the rhoptry bulb (Fig. 4a and b). For a more detailed view of PfCERLI2 and its association with the rhoptry, we imaged double labelled parasites in three-dimensions using the super-resolution microscopy platform Airyscan, which showed that PfCERLI2 forms the donut shape characteristic of membrane-associated rhoptry bulb proteins (Fig. 4c).

Repeat immunofluorescence experiments consistently showed that mature schizonts had little PfCERLI2 staining. Parallel western blots, however, showed no significant decrease in PfCERLI2 detection between early and late schizonts (Supplementary Fig. 10). Given the lack of change in detectability by western blot, it is possible that the HA epitope is masked and therefore undetectable using anti-HA antibodies in microscopy with mature schizont stage parasites.

**PfCERLI2 is peripherally associated with the cytosolic face of the rhoptry bulb membrane.** The absence of a signal peptide suggested that PfCERLI2 is a peripheral membrane protein that localises to the rhoptries but is exposed to the merozoite cytosol. To test this, we subjected PfCERLI2HAGlmS schizont lysates to a proteinase protection assay. Following lysis with saponin and digitonin, which lyse the RBC membrane, PVM, and PPM, but not organellar membranes, PfCERLI2 was sensitive to proteinase K degradation (Fig. 5a); suggesting it is exposed to the cytosol. To assess membrane association of PfCERLI2, we performed a solubility assay using lysates from PfCERLI2HAGlmS schizonts (Fig. 5b), whereby proteins are solubilised into cytosolic (hypotonic lysis), peripheral membrane (carbonate treatment), integral membrane (triton-x-100 soluble), or covalently lipid-linked (triton-x-100 insoluble) fractions. PfCERLI2 predominantly solubilised in the carbonate fraction, a distinctive feature of peripheral membrane proteins[26]. Combining the localisation analyses with the proteinase protection and solubility assays suggest that PfCERLI2 localises to the cytosolic face of the rhoptry bulb membrane, a subcellular localisation akin to that reported for PfCERLI1[13].

**PfCERLI2 knockdown inhibits rhoptry bulb antigen processing but not secretion of rhoptry neck proteins.** PfCERLI1 has previously been shown to be involved with rhoptry secretion[13,14], and given the similarities between PfCERLI1 and PfCERLI2 we sought to determine whether PfCERLI2 was also involved in rhoptry secretion. To test the effect of PfCERLI2 knockdown on rhoptry secretion, ring-stage PfCERLI2HAGlmS parasites were treated with 2.5 mM GLCN or left untreated. The RBCs of each culture had been treated with trypsin, chymotrypsin, and neuraminidase to prevent reinvasion. Parasites were cultured until schizont rupture when the supernatant was collected (containing secreted proteins) and a parasite lysate (containing merozoites that had failed to invade) were prepared for western blot (Fig. 6a). As expected, there was no change in secretion of the micronemal antigen erythrocyte binding antigen 175 (EBA-175) with knockdown of the rhoptry associated PfCERLI2 (Fig. 6b). There was also no noticeable defect in the secretion of the rhoptry neck antigens reticulocyte binding homologue 4 (RH4) and rhoptry neck protein 4 (RON4). These data indicate that PfCERLI2 knockdown does not block secretion of rhoptry neck antigens in merozoites that fail to invade enzyme-treated RBCs.

Parasite lysates from the secretion assay, containing free merozoites, were also probed with antibodies to the rhoptry bulb antigen rhoptry associated protein 1 (RAP1) (Fig. 6a). RAP1 has previously been shown to be processed by Plasmepsin IX and subtilisin-like protease 1 (SUB1) from an 84 kDa precursor in intact schizonts into a 67 kDa mature product in free

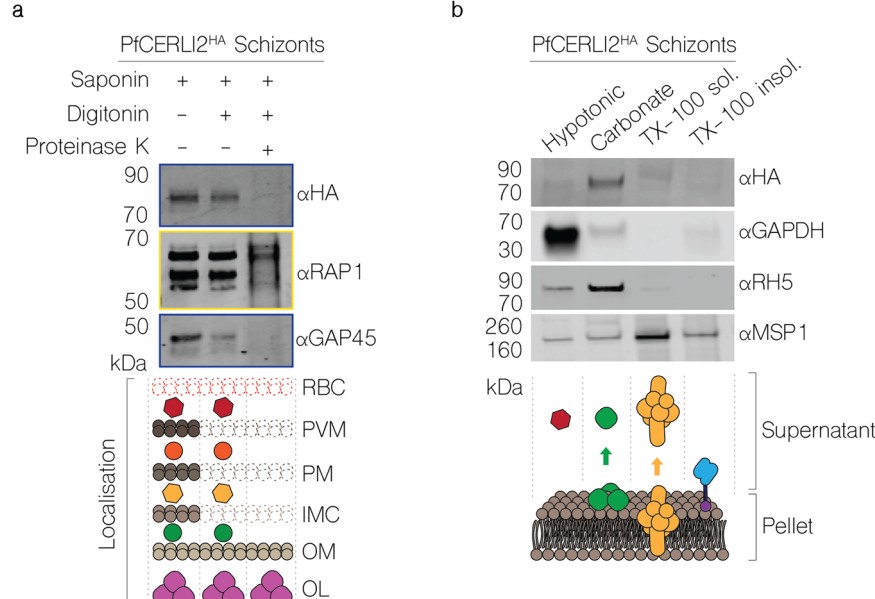

**Fig. 5 PfCERLI2 is peripherally associated with the cytosolic face of the rhoptry bulb membrane. a** PfCERLI2$^{HAGlmS}$ Schizont lysates were treated with either saponin alone, saponin and digitonin, or saponin, digitonin, and proteinase K. Saponin lyses the RBC membrane and PVM, while digitonin lyses the parasite plasma membrane but leaves the membrane of internal organelles intact. Treated lysates were then probed with a cocktail of anti-HA (PfCERLI2) and anti-GAP45 (inner-membrane complex, exposed to cytosol) antibodies (outlined in blue) or RAP1 (rhoptry bulb lumen) antibodies (outlined in yellow). Antibodies against RAP1 and GAP45 serving as negative and positive controls, respectively, for proteinase K digestion. Images representative of 3 independent experiments. RBC = RBC membrane, PVM = PV membrane, PM = parasite plasma membrane, IMC = inner-membrane complex, OM = organellar membrane, OL = organelle lumen. **b** PfCERLI2$^{HAGlmS}$ schizont lysates were subjected to differential lysis and solubilisation to determine the membrane solubility of PfCERLI2. Lysates were first hypotonically lysed, before being treated sequentially with sodium carbonate and triton-x-100. The supernatant was collected following each treatment, along with a final triton-x-100 soluble fraction. Each fraction was then probed with anti-HA (PfCERLI2), anti-GAPDH (cytosolic), anti-RH5 (peripheral) and anti-MSP1 (GPI-anchored) antibodies. Images representative of 3 independent experiments.

merozoites[27–30]. Diminished PfCERLI2 expression leads to the inhibition of RAP1 processing, corresponding to an accumulation of the unprocessed form of the protein (Fig. 6c). When PfCERLI2$^{HAGlmS}$ schizonts either with, or without 2.5 mM GLCN were arrested prior to parasitophorous vacuole membrane (PVM) rupture with the protein kinase G inhibitor compound 1 (C1)[31] and probed for RAP1 (Supplementary Fig. 11a), both 2.5 mM GLCN treated and untreated cultures showed mostly unprocessed RAP1, supporting that the defect in RAP1 processing occurs only after schizont rupture and merozoite release (Supplementary Fig. 11b). Collectively, this suggests that knockdown of PfCERLI2 interferes with the processing of RAP1 between PVM rupture and merozoite invasion, a phenotype previously demonstrated for PfCERLI1 knockdown[13].

**PfCERLI2 knockdown alters rhoptry morphology.** While PfCERLI2 knockdown schizonts showed no gross morphological defects, we aimed to determine if PfCERLI2 knockdown altered the morphology or biogenesis of the rhoptries. We matured GLCN treated or untreated 3D7 WT and PfCERLI2$^{HAGlmS}$ schizonts in the presence of C1 and subsequently fixed and imaged these mature schizonts using thin-section transmission electron microscopy (TEM) (Fig. 6d). PfCERLI2$^{HAGlmS}$ schizonts that had not been treated with GLCN were morphologically indistinguishable to both GLCN treated and untreated 3D7 WT parasites. By contrast, GLCN treated PfCERLI2$^{HAGlmS}$ parasites appeared to contain noticeably elongated rhoptry necks. We measured rhoptry length in randomly selected sections (38 untreated, 33 GLCN treated) and the measured rhoptry lengths were on average ~90 nm (±3.4% SEM) longer in GLCN treated

PfCERLI2$^{HAGlmS}$ parasites than those of untreated controls (Fig. 6e).

Dimensional analysis of organelles is partially compromised in thin section TEM, as XY measurements of an organelle are influenced by differences in sectioning profile. To overcome this, we performed array tomography of both GLCN treated and untreated PfCERLI2$^{HAGlmS}$ schizonts (Fig. 7a–f) to analyse 3-D changes in rhoptry morphology. Rhoptries of GLCN treated PfCERLI2$^{HAGlmS}$ parasites ($n = 100$) were on average 130 nm (±10 nm SEM) (Fig. 7g) longer than rhoptries from untreated parasites ($n = 100$) representing a 24.17% rhoptry lengthening following PfCERLI2 knockdown. In addition, rhoptries of GLCN treated PfCERLI2$^{HAGlmS}$ parasites exhibited similar increases in surface area (Fig. 7h) and volume (Supplementary Fig. 12a) but no change in surface area to volume ratio (Supplementary Fig. 12b). By contrast to their lengthening, GLCN treated PfCERLI2$^{HAGlmS}$ parasites showed a small decrease in rhoptry width (Supplementary Fig. 12c) by an average of 12 nm (±5.5 nm SEM), leading to overall longer and narrower rhoptries following PfCERLI2 knockdown (Supplementary Fig. 12d).

To further confirm the rhoptry morphological changes we observed by electron microscopy, we performed super-resolution (airyscan) microscopy on either untreated or 2.5 mM GLCN treated PfCERLI2$^{HAGlmS}$ parasites, probed with rhoptry bulb and neck markers RAP1 and RON4. We then measured the distance between the basal end of the nucleus and the centre of the RAP1 or RON4 (Supplementary Fig. 13) foci, determining the average distance between nucleus and rhoptry. In GLCN treated parasites, we consistently observed a shorter distance between the nucleus and both RAP1 (Fig. 7i) and RON4 (Fig. 7j) foci (Supplementary Fig. 13). This closer proximity of rhoptry markers to the nucleus

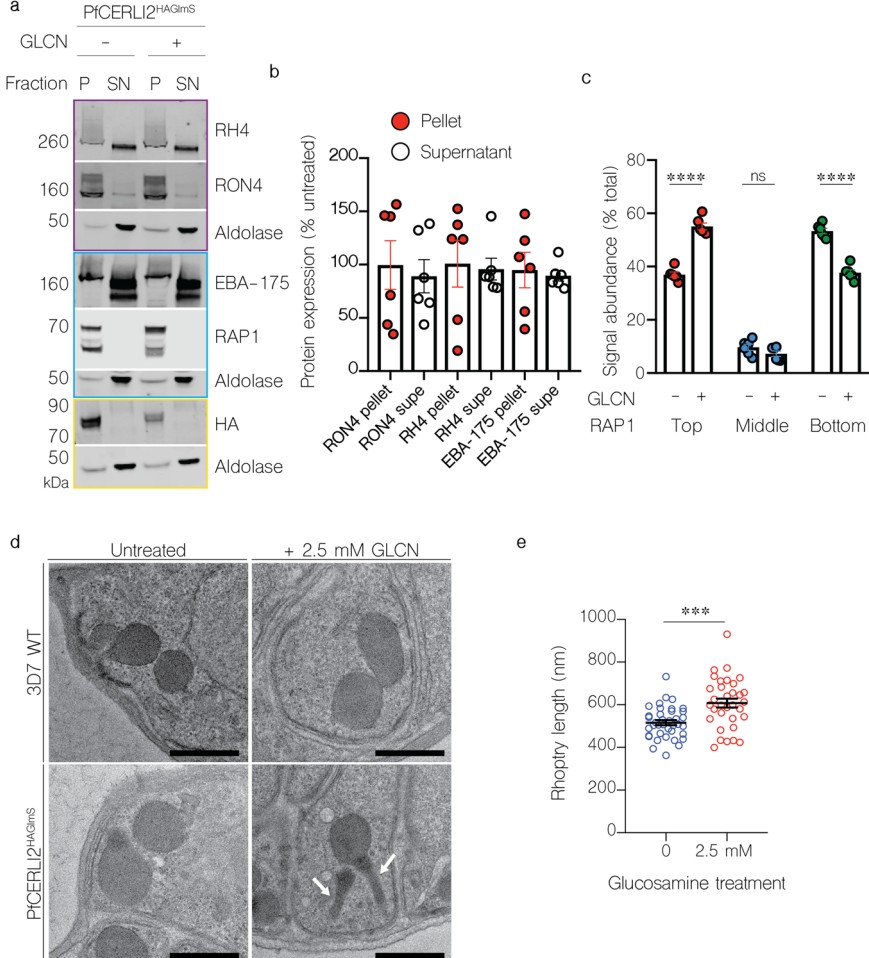

**Fig. 6 Knockdown of PfCERLI2 inhibits rhoptry antigen processing, but not secretion, and is associated with rhoptry malformation. a** PfCERLI2$^{HAGlmS}$ ring-stage parasites were GLCN treated (+) or left untreated (−), before enzyme treatment of RBCs to prevent reinvasion. Lysates (P) and culture supernatants (SN) were harvested following schizont rupture and probed with antibody cocktails according to the colour theme: anti-RH4, anti-RON4 (rhoptry neck), anti-aldolase (loading control) antibodies (outlined in purple), anti-EBA-175 (microneme), anti-RAP1 (rhoptry bulb), anti-aldolase (loading control) antibodies (outlined in blue), anti-HA (PfCERLI2), anti-aldolase (loading control) antibodies (outlined in yellow). Representative blots of 5 biological replicates shown. **b** Western blots were normalised to the loading control, aldolase, and quantified, with results displayed as protein expression in GLCN treated sample as a percentage of the untreated signal. $n = 5$ biological replicates. **c** Quantification of individual band intensities, as a percentage of the total signal, from RAP1 signals using the parasite lysates from the secretion assay. $n = 5$ biological replicates. ns $= p > 0.05$, ****$p < 0.0001$ by two-way ANOVA. **d** Transmission electron microscopy of C1 schizont arrested PfCERLI2 and 3D7 WT parasites, either in the presence or absence of GLCN. Representative images from 3 biological replicates. Scale bar = 500 nm. **e** Quantification of rhoptry length from transmission electron microscopy images. $n = 3$ biological replicates. ***$p < 0.001$ by unpaired $t$-test. All error bars = SEM.

supports the observation of rhoptry lengthening following PfCERLI2 knockdown.

To determine if distribution of these rhoptry markers changed following PfCERLI2 knockdown, we also quantified the area, volume, and sphericity of RAP1 and RON4 foci in both untreated and 2.5 mM GLCN treated PfCERLI2$^{HAGlmS}$ parasites (Supplementary Fig. 14). Consistent with the observation of an elongated rhoptry neck following PfCERLI2 knockdown, RON4 foci of GLCN treated parasites displayed a much larger surface area (Supplementary Fig. 14a), larger volume (Supplementary Fig. 14b) and were less spherical (Supplementary Fig. 14c). In addition, RAP1 foci in 2.5 mM GLCN treated parasites displayed a lower surface area and volume (Supplementary Fig. 14d and e), although the magnitude of this change was small, but no change in sphericity (Supplementary Fig. 14f). To determine if these minor changes in RAP1 foci shape following PfCERLI2 knockdown were indicative of a change to the size of the rhoptry bulb, RAP1 foci diameter was measured; but no change in rhoptry bulb

diameter was observed in GLCN treated parasites (Supplementary Fig. 14g). Thus, PfCERLI2 knockdown alters rhoptry morphology and is associated with incomplete processing of the rhoptry bulb protein RAP1, suggesting that PfCERLI2 plays an important role in normal rhoptry physiology.

**Proximity-dependent biotin interaction studies fail to show an interaction between PfCERLI1 and PfCERLI2.** Given that both PfCERLI1 and PfCERLI2 share localisation on the cytosolic face of the rhoptry bulb membrane and exhibit several similar knockdown phenotypes, we sought to determine whether the two proteins interacted as a complex in this niche. To assay for an interaction, we utilised dimerization-induced quantitative proximity-dependent biotin identification (DiQ-BioID), a recently developed method in the field (Fig. 8a)[32], and generated transgenic parasite lines, *Pfcerli1*$^{DiQ-BioID}$ and *Pfcerli2*$^{DiQ-BioID}$, which each separately expressed the bait PfCERLI1 and PfCERLI2

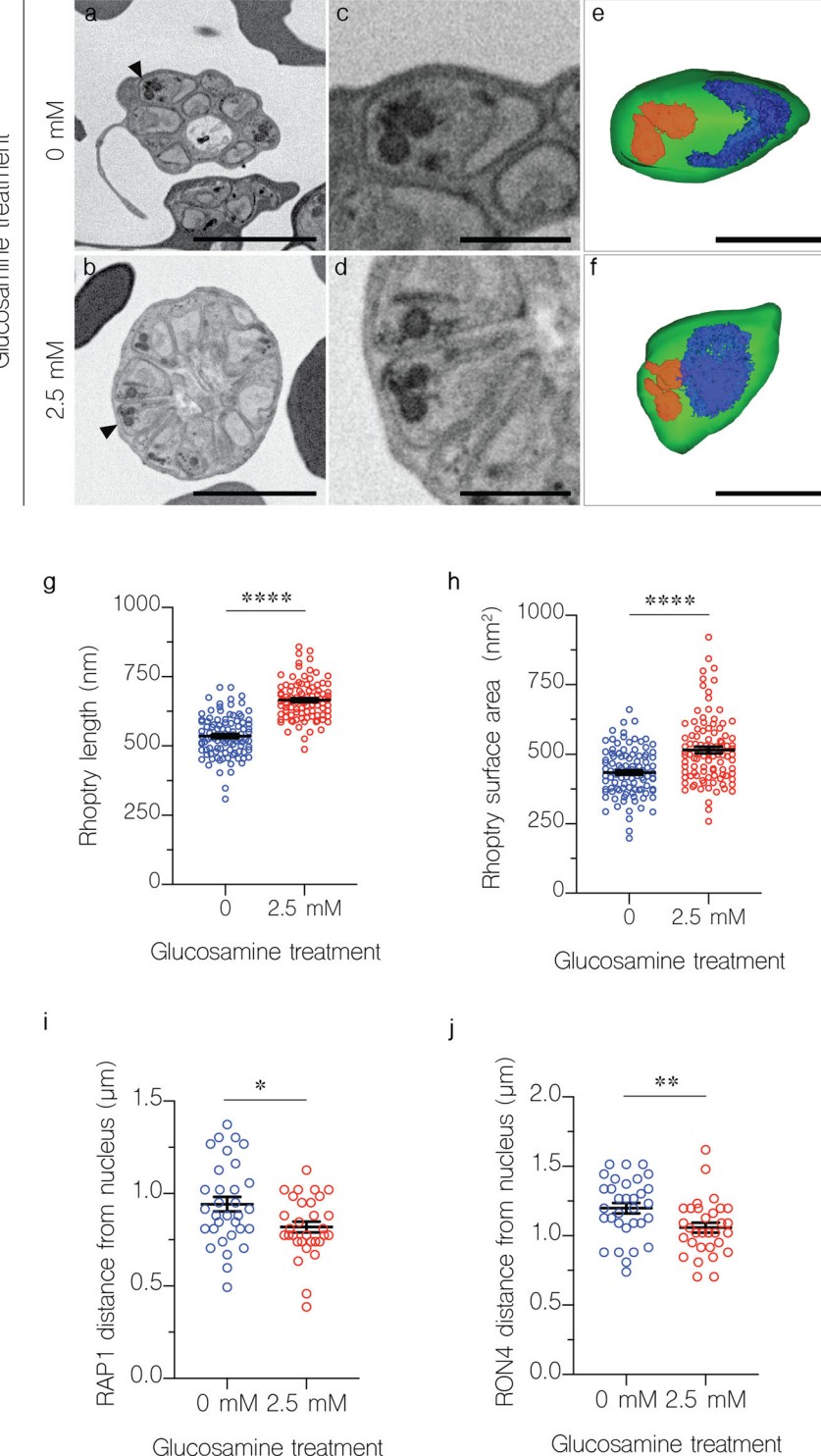

**Fig. 7 Knockdown of PfCERLI2 alters rhoptry length and positioning.** PfCERLI2[HAGlmS] schizonts were matured in the presence of E64 and imaged using array tomography. Inverted backscattered images of untreated (**a**) and GLCN treated (**b**). PfCERLI2[HAGlmS] schizonts. **c**, **d** Higher magnification images revealing individual merozoites and associated rhoptry pairs indicated by black arrow heads in (**a**) and (**b**). **e**, **f** 3D renderings corresponding to the images in (**c**) and (**d**). Colour legend: Merozoite plasma membrane, green; nucleus, blue; rhoptries, red. Scale bars: **a** and **b** = 4 µm; **c**–**f** = 1 µm. Using array tomography, 100 rhoptries for each treatment were analysed to determine their length (**g**) and surface area (**h**). PfCERLI2[HAGlmS] schizonts were matured in the presence of E64, stained with antibodies against RAP1 (rhoptry bulb) and RON4 (rhoptry neck), and imaged by Airyscan super-resolution microscopy. The flourescence intensity from maximum-intensity projections of RAP1 (**i**) and RON4 (**j**) signals were then measured from the basal end of the nucleus. 32 merozoites for untreated and 31 merozoites for 2.5 mM GLCN treated were measured from a total of 12 schizont images across three biological replicates. Representative images corresponding to this data in Supplementary Fig. 13. Error bars = SEM. *$p < 0.05$, **$p < 0.01$, ****$p < 0.0001$ by unpaired $t$-test.

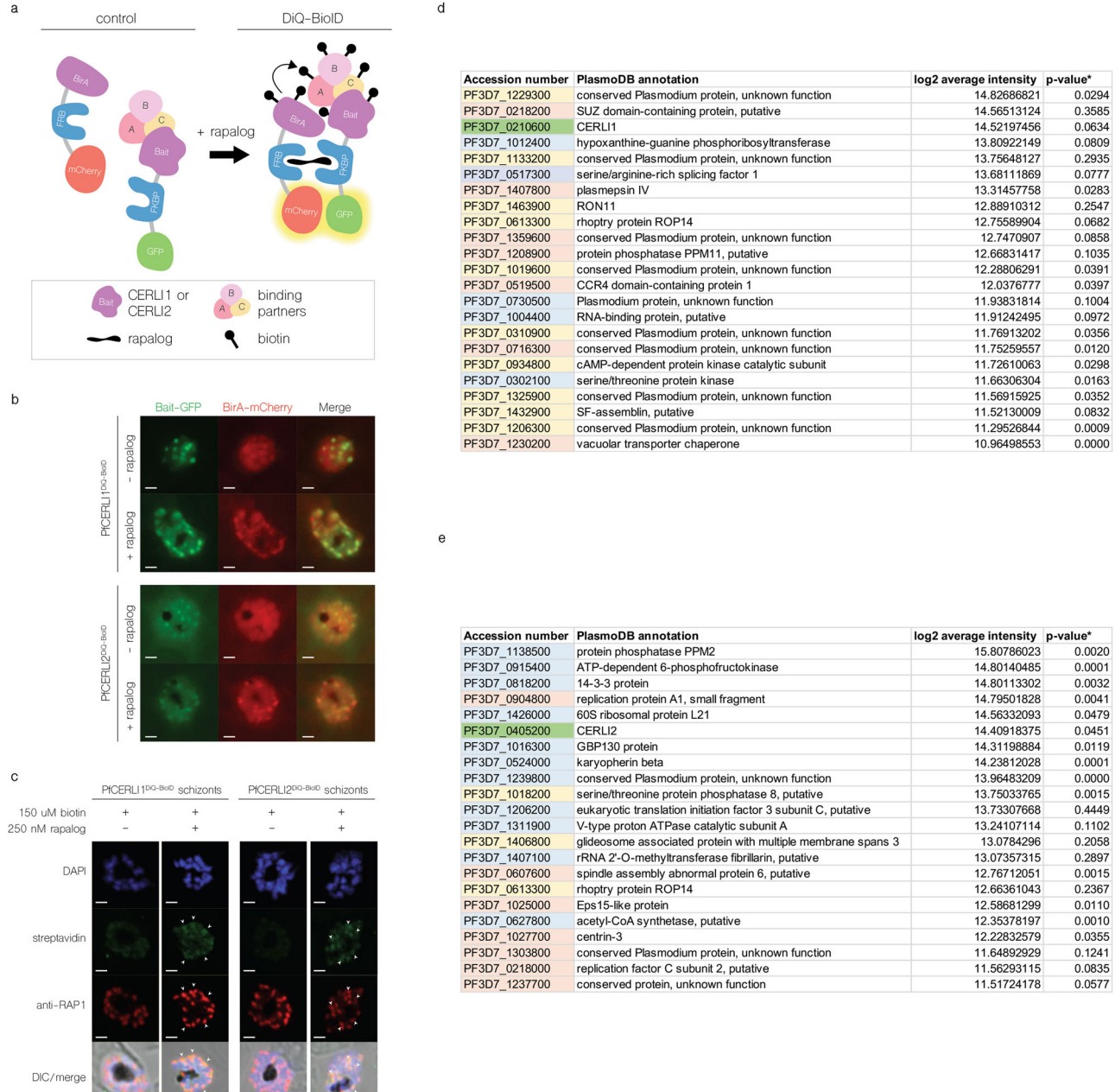

**Fig. 8 DiQ-BioID of PfCERLI1 and PfCERLI2 reveals their protein interactome. a** A schematic of the DiQ-BioID system under control and test conditions. **b** *Pfcerli1*^DiQ-BioID and *Pfcerli2*^DiQBioID parasites were treated for 8 h on rapalog to permit for dimerisation between the GFP-tagged bait construct and mCherry-tagged BirA, visualised by live fluorescence. Scale bars = 2 μm. **c** Biotinylation at the rhoptry bulb was confirmed by co-localising foci between RAP1 staining and a streptavidin-conjugated AlexaFluor488 in fixed cells by IFA. Scale bars = 2 μm. **d** Protein interactors for PfCERLI1. N = 3 biological replicates. **e** Protein interactors for PfCERLI2. N = 3 biological replicates. **d, e** Colour code corresponds to: green (bait protein), yellow (peak schizont-expressed), pink (late trophozoite to schizont stage-expressed), blue (uniformly or ring stage-expressed). *P-value calculated by t-test between intensity values in double-treated parasites and no rapalog controls. Proteins not identified in the no rapalog control were assigned with an intensity value of 0.

tagged with an FK binding protein (FKBP) and GFP. These lines are then co-transfected with an episomally-expressed BirA biotin ligase fused to an FKBP rapamycin-binding protein domain (FRB) and mCherry (Supplementary Fig. 15a). Integration or episomal maintenance of these constructs were confirmed by PCR (Supplementary Fig. 15b).

Upon the addition of rapalog, the bait and BirA are able to dimerise, and the system can then be validated by live fluorescence of co-localising GFP/mCherry foci (Fig. 8b). BirA, in the presence of excess biotin, then proceeds to biotinylate proteins in close proximity to the FKBP tagged bait protein,

which we verified in fixed-cell IFA using a streptavidin-conjugated fluorophore signal co-localising with RAP1 (Fig. 8c). To prepare samples for proteomics, 36-h-old parasites were double-treated with rapalog and biotin for 8 h, including 4 h on E64, before being processed for streptavidin-based pulldown and mass spectrometry analysis.

To minimise the discovery of false positives from non-specific biotinylation, hits were first normalised against samples without rapalog treatment and then to a BirA-control line that has no FKBP tagged bait protein. A representative western blot for the extent of biotinylation between test and control samples are

included in Supplementary Fig. 15c. In order to identify proteins that were reliably enriched, we only considered hits which were identified with at least 2 peptides with an overall peptide coverage of 10% or more from the in-house *P. falciparum* library for all biological replicates (replicate comparison Supplementary Fig. 16). Applying these criteria generated an interactome of 23 proteins for PfCERLI1 (Fig. 8d), and 26 proteins for PfCERLI2 (Fig. 8e), which included the bait proteins themselves. PfCERLI2 was not identified as an interactor for PfCERLI1 and vice versa. Nevertheless, several rhoptry and schizont-expressed proteins were identified for both datasets to highlight the target specificity of DiQ-BioID. Of greatest interest includes RON11, ROP14 and the cAMP-dependent catalytic subunit of protein kinase A (PKAc) as potential interactors of PfCERLI1. Likewise, PfCERLI2 appears to interact with ROP14. Both lists display several uncharacterised proteins of unknown function that have peak expression at schizogony.

## Discussion

We and Suarez et al. had previously characterised the rhoptry bulb protein PfCERLI1 and determined that it played an essential role in merozoite invasion and rhoptry secretion[13,14]. Bioinformatic searches identified PfCERLI2 as a protein potentially related to PfCERLI1 based on their amino acid sequence homology. Here, we show that *cerli2* likely arose through ancestral gene duplication from *cerli1*. Furthermore, we show that like PfCERLI1, PfCERLI2 also localises to the cytosolic face of the rhoptry bulb, is essential for merozoite invasion and has important functions in rhoptry biology.

PfCERLI2 localised to the rhoptry bulb by immunofluorescence assays and solubilised mainly in the carbonate treatment, suggesting it is peripherally associated with the rhoptry bulb membrane. PfCERLI1 was present in both the carbonate and triton-x100 insoluble fractions, which we hypothesised may be a result of strong cooperative membrane association through its C2 domain, PH domain, and palmitoylation site[13]. By contrast, PfCERLI2 appears to lack a functional PH domain and the palmitoylation site possessed by PfCERLI1. In the most likely *Toxoplasma gondii* homologue of PfCERLI2 (TGME49_315160, also known as TgRASP2), the PH domain is involved in, but not essential for, rhoptry localisation and secretion[14]. Based on evidence from its *T. gondii* homologue and the lack of a canonical PH domain and palmitoylation site, we suggest that membrane attachment of PfCERLI2 is likely mediated by its C2 domain, a structure involved in targeting many proteins to various membranes[33].

Knockdown of PfCERLI2 led to reduced parasite growth and merozoite invasion. Morphology and rate of schizont rupture in PfCERLI2 knockdown parasites were indistinguishable compared to the controls, indicating merozoite development and egress was not impacted by knockdown of PfCERLI2. Instead, we observed increased numbers of both free merozoites in the culture medium and merozoites bound to the RBC surface in PfCERLI2 knockdown parasites. Based on this evidence, the inducible growth inhibition seen with PfCERLI2 knockdown is most likely to be a result of merozoite invasion being interrupted at or before formation of the irreversible tight junction with the RBC surface. This is further supported by a decrease in AMA1 diameter in merozoites bound to the RBC for PfCERLI2 knockdown parasites, suggesting that the invasion process had been halted at the early stages of host-cell entry and tight junction formation. Indeed, a corresponding increase in RBC bound merozoites following PfCERLI2 knockdown suggests that rhoptry neck and micronemal protein contents were secreted for PfCERLI2 deficient merozoites, and the tight junction formed for some of these,

but these merozoites were then unable to proceed with invasion. The fact that we saw no reduction in secretion of the rhoptry neck proteins RON4 and RH4 following CERLI2 knockdown supports the idea that in many cases the glucosamine inducible ribozyme PfCERLI2 knockdown parasites can secrete enough protein to form the tight junction, but these merozoites fail to invade due to a later-occurring defect.

The presence of a prominent decapeptide repeat region at the C-terminus of PfCERLI2 is a differentiating feature compared to PfCERLI1. Based on its differential evolution relative to the rest of the PfCERLI2 protein, and its exclusive presence in *Laverania*, we hypothesise that the decapeptide tandem repeat number may be under positive selection, whereas the rest of the protein is under negative selection that strongly favours amino acid conservation across *Plasmodium* spp. A previous study identified a growth inhibitory monoclonal antibody (known as M26-32[18]) that bound to PfCERLI2, as well as a number of other proteins[34], and was reported to localise this protein to the merozoite surface. This study also reported that the decapeptide tandem repeat contained highly immunogenic antigens that bind malaria-exposed serum antibodies. A potential mechanism for the positive selection of the PfCERLI2 repeat could therefore be classical antibody selection. However, our data using endogenously tagged PfCERLI2 shows that the protein localises to the cytosolic face of the rhoptry bulb membrane and has a role in the function of this organelle, making it unlikely that PfCERLI2 is exposed to the host immune system and antibody selection. We therefore suggest that the PfCERLI2 repeat has an as yet unidentified function in *P. falciparum* and its evolution is under positive selection based on its Ka/Ks value of 1.43 across *Laverania*. Amongst *P. falciparum* isolates, repeat number varies and it is currently unclear whether the same occurs among different isolates of all *Laverania*, or whether this is a unique property of PfCERLI2. Given the conservation of the repeat sequence in *P. falciparum*, but the variation in repeat number, this variation in repeat number may have an important role in this protein's function between isolates.

Due to the presence of two paralogous genes in all coccidians (e.g. *Toxoplasma gondii* and *Besnoitia* spp.) and haematozoa (e.g. *Plasmodium* spp.), it is likely that an ancestral *cerli1* sequence was duplicated in the most recent common ancestor of these related parasite lineages. By contrast, all more divergent apicomplexans and Chromerids contain only a single *cerli* gene, suggesting that there was originally only a single *cerli* gene in the more distant ancestors of coccidia and haematozoa. *Cryptosporidium* is more divergent than coccidia and haematozoa and contains only a single CERLI homologue. As the gene structure of the *cerli* homologue of *Cryptosporidium* is highly similar to that of *cerli1* across other Apicomplexa, we suggest that *cerli1* is most similar to the ancestral gene that underwent duplication and gave rise to the more divergent *cerli2* gene. From studies of both PfCERLI1 and PfCERLI2, it is clear that both proteins have functional similarities due to their stage of expression and localisation, along with their knockout and knockdown phenotypes[13,14,16]. In addition, in *Toxoplasma gondii* it has been reported that TgRASP1 and TgRASP2 form a complex[14]. Our DiQ-BioID data for PfCERLI1 and PfCERLI2 currently does not support a direct interaction between these proteins in *P. falciparum*. However, it is still possible that the proteins interact via another protein or a complex that may not be detected by DiQ-BioID in intact cells but can be immunoprecipitated in solubilised cells using a HA-tag. Low expression of PfCERLI2 is another possible explanation. For the coccidia and haematozoa containing *cerli* paralogues, it is likely that the duplicated *cerli2* locus that arose from gene duplication of an ancestral *cerli1* locus underwent subfunctionalisation. This is supported by shared characteristics of PfCERLI1 and PfCERLI2, along with the strong negative selection

CERLI2 has been under across Apicomplexa and the presence of only a putative remnant of a PH domain.

Dimerization-induced proximity-based biotinylation using either PfCERLI1 or PfCERLI2 as the bait delivered a list of putative interacting proteins. The most notable one is the multi-transmembrane domain-containing ROP14, which appears in the interactomes for both proteins. Though the function of ROP14 remains largely unknown, it has been localised in *P. falciparum* schizonts by IFA and denoted to be slightly posterior to RON4[35] and therefore may possess a rhoptry bulb localisation. It is possible that ROP14, in complex with other proteins, forms either a bridge between PfCERLI1 and PfCERLI2 that was not detected within the biotinylation range of BirA, which can vary from 10 to 30 nm[36]. Alternatively, ROP14 may transiently interact with both proteins, but not simultaneously.

CERLI2 homologues in coccidia and haematozoa possess a PH domain towards their C-terminus that is under negative selection. The exception to this is the genus *Plasmodium*, where the region corresponding to the PH domain of PfCERLI1 in CERLI2 has undergone significant change in sequence and structure such that a PH domain is no longer predicted for, or likely functional in, this region of *Plasmodium* spp. CERLI2. In addition, PfCERLI2 homologues in the *Laverania* subgenus of *Plasmodium* also have a decapeptide tandem repeat that is under positive selection. While it is not clear what the function of this repeat region is, the likely absence of the PH domain's lipid-binding function[14] and presence of the decapeptide repeat region may indicate that CERLI2 in *Laverania* has undergone neosubfunctionalisation, where following subfunctionalisation one paralogue adopts a new function[37]. We can therefore speculate that CERLI2 homologues in *Laverania* may be evolving modified functions or different protein-protein/protein-lipid interactions compared to homologues outside this subgenus.

This study highlights that CERLI2, like its paralogue CERLI1, has an important role in *Plasmodium* invasion and rhoptry biology that is likely shared broadly across Apicomplexa. In *P. falciparum*, loss of PfCERLI2 led to elongated rhoptry structures that were evident in both 2D and 3D cell analysis and was associated with reduced merozoite invasion. The localisation of PfCERLI2 puts it in the right place at the right time to undertake overlapping functions with its paralogue PfCERLI1 on the cytosolic face of the rhoptry bulb, and this possibility may be resolved in future studies. The modified structure of CERLI2 in *Plasmodium* spp. suggests that the requirements of malaria parasite host-cell invasion has led to further evolution of the protein, highlighting the ongoing specialisation of these proteins as the apicomplexan parasites diverged.

## Methods

**Bioinformatic analyses**. PfCERLI2 (Pf3D7_0405200), PfCERLI1 (Pf3D7_0210600) and orthologous sequences used for comparison *of Pfcerli1* and *Pfcerli2* gene structure from *Babesia bigemina* (BBBOND_0209090, BBBOND_0311360), *B. bovis* (BBOV_IV005720, BBOV_III003780), *Cryptosporidium andersoni* (cand_006180), *C. hominis* (Chro.60252), *C. muris* (CMU_007720), *C. parvum* (cgd6_2143), *C. tyzzeri* (CTYZ_00001395), *C. ubiquitum* (cubi_02338), *Chromera Velia* (Cvel_21914), *Cyclospora Cayetanensis* (cyc_01372), *Cystoisopora Suis* (CSUI_005102), *Cytauxzoon Felis* (CF002482, CF003293), *Eimeria brunetti* (EBH_0041390), *G. niphandrodes* (GNI_053280), *N. caninum* (NCLIV_057920, NCLIV_049720), *P. reichenowi* (PRCDC_0402800, PRCDC_0209500), *P. malariae* (PmUG01_03015300, PmUG01_04021700), *P. ovale* (PocGH01_03012700, PocGH01_04019500), *P. knowlesi* (PKNH_0303300, PKNH_0410600), *P. vivax* (PVP01_0304600, PVP01_0414300), *P. berghei* (PBANKA_1002900, PBANKA_0307500), *P. yoelii* (PY17X_1004300, PY17X_0308100) and *P. chabaudi* (PCHAS_1003800, PCHAS_0309700), *S. neurona* (SN3_01800355, SN3_00600405), *T. annulata* (TA21020, TA12010), *T. equi* (BEWA_030720, BEWA_019330), *T. gondii* (TGME49_315160, TGME49_235130), *T. orientalis* (TOT_010000334, TOT_020000207), *V. brassicaformis* (Vbra_14966) were obtained by searching within the EuPathDB.org database[38]. In addition, homologues of PfCERLI2 from the Laveranian parasites *P.*

*adleri* (PADL01_0404800), *P. billcollinsi* (PBILCG01_0404100), *P. blacklocki* (PBLACG01_0402700), *P. gaboni* (PGABG01_0403400) and *P. praefalciparum* (PFRFG01_0404700) were obtained by searching within the PlasmoDB.org database[15].

Both pairwise and multiple sequence alignments were generated using Geneious 9.1.3 (Biomatters) using a Geneious global alignment with free end gaps (gap open penalty = 12, gap extension penalty = 3) algorithm with the Blosum62 cost matrix. Multiple sequence alignments were also constructed using Muscle[39] in an R environment[40]. Phylogenetic trees were constructed from nucleotide and amino acid sequences using the Jukes-Cantor[41] DNA or LG[42] amino acid substitution models, respectively, genetic distance model and were built using the unweighted pair group method with arithmetic mean (UPGMA) tree build method using Geneious or the R package phangorn v. 2.5.5[43]. A tanglegram visualisation was generated using the cophyloplot function of the R package ape[44].

To identify *P. falciparum* proteins that contained a PHIS, or PHIS-like motif, the PHIS sequence from PfCERLI1, PfCERLI2, or the consensus sequence generated from their apicomplexan homologues was used in a BLASTp search against the *P. falciparum* 3D7 proteome.

Ka/Ks ratios were calculated using the online Ka/Ks Calculation tool (http://services.cbu.uib.no/tools/kaks)[45,46] using the maximum likelihood tree method, discrete_Grantham submatrix, and a LI rate of moderate. Where Ka/Ks ratios are reported, they were generated using the same coding sequences as the corresponding phylogenetic tree.

**Protein structure prediction**. The protein structure of PfCERLI1, PfCERLI2, along with their homologues in *T. gondii* (TGME49_235130, TGME49_31516) and *C. muris* (CMU_007720) was predicted using the online protein structure prediction tool Phyre2 on intensive mode[47]. All predicted structures were visualised, presented and imaged using either Jmol[48] or EzMol[49].

**Continuous culture of asexual stage *P. falciparum***. *P. falciparum* 3D7 parasites were cultured in O$^+$ red blood cells (Red Cross Blood Service Australia) as previously described[50]. Parasites were grown in RPMI-HEPES media (Sigma-Aldrich) supplemented with 0.5% v/v Albumax (Gibco), 52 mM gentamycin (Gibco), 367 mM hypoxanthine (Sigma-Aldrich), 2 mM L-Glutamax (Gibco) and 2 mM sodium bicarbonate (ThermoFisher Scientific), adjusted to a pH of 7.4. Cultures were maintained at 37 °C in sealed acrylic boxes with the following gas composition: 1% $O_2$, 5% $CO_2$ and 94% $N_2$.

**Plasmid construction and transfection**. List of genetic constructs in this study: *Pfcerli2*$^{HAGlmS}$, *Pfcerli2*$^{HAGlmS/GFP}$, *Pfcerli2*$^{SLI-TGD}$.

*Pfcerli2*$^{HAGlmS}$ riboswitch transfection vectors were prepared from a PTEX150$^{HAGlmS}$ plasmid backbone[51]. A 758 base pair homology region of *Pfcerli2* was PCR amplified from the 3′ end of the genomic sequence of 3D7 gDNA using the primers *Pfcerli2* 5′ F RBW (GGT**AGATCT**GAAGGAATCTTTTGGAGATGAGC) and *Pfcerli2* 3′ R RBW (GGT**CTGCAGC**TATATTTTGTATGGTATTTTCTAATTGTGC). Each of the resulting PCR products was digested with the restriction enzymes BglII and PstI (indicated in bold in primer sequence) and cloned into the PTEX150$^{HAGlmS}$ vector.

To generate the *Pfcerli2*$^{HAGlmS/GFP}$ parasite line, which expresses cytosolic GFP, the pHGBrHrBl-1/2 GFP plasmid was used, without modification[52].

For disruption of *Pfcerli2* using SLI-TGD system, a source SLI-TGD vector (Pf3D7_1463000 SLI-TGD) was used[19]. A 772 bp homology region of *Pfcerli2* was PCR amplified using the primers *Pfcerli2* SLI-TGD F (GGT**GCGGCCGC**TCCACATATAAGTGATTTCGAGCC) and *Pfcerli2* SLI-TGD R (GGT**ACGCGT**CCTTGCACACTTCTTGTCC). Both PCR product, and the SLI-TGD vector were then digested with NotI and MluI (restriction sites in bold) and ligated together.

To generate the *Pfcerli1*$^{DiQ-BioID}$ and *Pfcerli2*$^{DiQ-BioID}$ lines, we utilized pBirA-7xGGGGS-FRB-mCherry and pSLI-2xFKBP-GFP plasmids, the latter of which was modified with a homology flank for each gene[19,32]. Single homology flanks for each gene at their 3′ end were PCR amplified from NF54 gDNA: 767 base pairs for *Pfcerli1* using the primers *Pfcerli1* DiQ-BioID F (GGT**GCGGCCGC**CATAT CAAATTTGGTTCTTGAAG) and *Pfcerli1* DiQ-BioID R (GGT**CCTAGG**CAT CACTATAGTTGTACATATTTTGC), and 758 base pairs for *Pfcerli2* using *Pfcerli2* DiQ-BioID F (GGT**GCGGCCGG**GAAGGAATCTTTTGGAGATGAGC) and *Pfcerli2* DiQ-BioID R (GGT**CCTAGG**TATATTTTGTATGGTATTTT CTAATTGTGC). Each PCR product was digested with the restriction enzymes NotI and AvrII (sites indicated in bold) and cloned into pSLI-2xFKBP-GFP.

*P. falciparum* 3D7 or NF54 parasites were transfected by RBC loading[53] (*Pfcerli2*$^{HAGlmS}$, *Pfcerli2*$^{HAGlmS/GFP}$) or schizont transfection[54] (*Pfcerli2*$^{SLI-TGD}$, *Pfcerli1*$^{DiQ-BioID}$ and *Pfcerli2*$^{DiQ-BioID}$). For RBC loading, uninfected RBCs were centrifuged at 440 rcf for 1 min, washed in culture media and then washed in cytomix (0.895% KCl, 0.0017% CaCl$_2$, 0.076% EGTA, 0.102% MgCl$_2$, 0.0871% K$_2$HPO$_4$, 0.068% KH$_2$PO$_4$, 0.708% HEPES). Washed RBCs were then resuspended in cytomix containing 200 μg of the plasmid of interest. The combined RBCs and DNA were transferred to a 0.2 cm cuvette (Bio-Rad) to be electroporated (Bio-Rad) at 0.31 kV with a capacitance of 960 μF. Electroporated RBCs were rinsed from the

cuvette with culture media, centrifuged at 440 rcf for 1 min, and washed twice before being introduced to cultures of gelatin-purified schizont stage parasites. Alternatively for schizont transfection, late-stage schizont cultures were Percoll-purified[55] and resuspended in 90 μl transfection buffer (90 mM $Na_2HPO_4$; 5 mM KCl; 50 mM HEPES; 0.15 mM $CaCl_2$; pH 7.3) plus 10 μl of TE containing 50 μg of the plasmid of interest, and electroporated using a 2D Nucleofector (Lonza) on program U-033. Electroporated parasites were transferred into 1.5 mL tubes with 1 mL culture media and 200 μL uninfected RBCs and shaken at 37 °C for 30 min to promote merozoites to invade, before being placed back into culture. To generate *Pfcerli2*[HAGlmS], 5 nM WR99210 (Jacobus Pharmaceuticals) drug treatment was used to select for integration of the HA-*GlmS* construct over 3 cycles. For *Pfcerli2*[HAGlmS/GFP], the HA-*GlmS* line was transfected with the pHGBrHrBl-1/2 construct in order to episomally express cytosolic GFP then selected for using 5 μg/mL blasticidin-S-deaminase HCl (Merck Millipore). To generate *Pfcerli2*[SLI-TGD], maintenance of the SLI-TGD construct was selected using 4 nM WR99210 and integrants were selected using 400 μg/mL G418 (geneticin, Sigma). *Pfcerli2*[SLI-TGD] plasmids were transfected twice into 3D7 parasite culture, which after they grew on WR99210 and were selected each four times independently with G418 for 8 weeks.

**Assessment of in vitro blood-stage growth and invasion**. To investigate the effects of *Pfcerli2* knockdown on parasite growth 3D7 and *Pfcerli2*[HAGlmS] cultures were synchronised to ring-stages using sorbitol lysis and adjusted to 1% parasitaemia at 1% haematocrit in 96-well U-bottom plates (Corning) as previously described[56]. From this adjusted culture, 45 mL was then suspended in 5 mL of a 10 x stock of D-(+)-glucosamine hydrochloride (hereby referred to as GLCN, Sigma-Aldrich) or media to a final volume of 50 mL. Plates were then incubated in standard culture conditions for 72 h, until trophozoite stages the following cycle. Completed assays were then stained in 10 mg/mL ethidium bromide (Bio-Rad) in PBS and the final parasitaemia was assessed using the BD Accuri™ C6 Plus flow cytometer (PE-H, FSC-H filters for growth, PE-H, FITC-H for invasion to visualise GFP-fluorescent ring stages). All flow cytometry files were analysed using FlowJo Version 10 (Tree Star). Representative flow plots are included in Supplementary Fig. 17 to show our gating strategies.

To investigate the effect of *Pfcerli2* knockdown on invasion, *Pfcerli2*[HAGlmS] cultures were synchronised to early trophozoite stages using sorbitol lysis and adjusted to 1% parasitaemia at 1% haematocrit and placed in 96-well U-bottom plates and set up as described for growth assays. Plates were subsequently incubated in standard culture conditions for 24 h, until early ring stages the following cycle. Completed assays were stained with ethidium bromide and analysed by flow cytometry as described for growth assays.

To observe whether PfCERLI2 knockdown had an impact on merozoite quantity per schizont or merozoite morphology, *Pfcerli2*[HAGlmS] parasites were treated with or without 2.5 mM GLCN at ring stages until matured into 40 h schizonts. To ensure full merozoite development, schizonts were arrested with E64 for 5 h to prevent rupture and then smeared as a thin blood film. Smears were fixed in 100% v/v methanol and stained in 10% v/v Giemsa (Merck Millipore) in water and then blindly assessed by light microscopy ($n = 20$ individual schizonts). To count free merozoites, *Pfcerli2*[HAGlmS/GFP] schizonts +/− 2.5 mM GLCN without E64 were permitted to incubate until rupture and gated for merozoites by flow cytometry as described above.

**Schizont rupture assay**. To observe whether PfCERLI2 knockdown affected schizont rupture and merozoite egress, synchronous *Pfcerli2*[HAGlmS] ring stages were treated in duplicate with or without 2.5 mM GLCN in 96-well U bottom plates as described above. Parasites were permitted to grow into schizonts prior to rupture and the parasitaemia was recorded by flow cytometry. Plates were then incubated for an additional 6 h to allow for schizont rupture followed by a second round of parasitaemia measurement by flow cytometry. To calculate the percentage of schizont rupture within this time window, the following Eq. (1) was used:

$$\% \text{ schizont rupture} = \left( \frac{\text{post} - \text{rupture schizontaemia}}{\text{pre} - \text{rupture schizontaemia}} \right) \times 100 \quad (1)$$

**Bound merozoite assay by light microscopy**. To determine whether PfCERLI2 knockdown inhibited merozoite invasion prior to, or following, tight junction formation PfCERLI2[HAGlmS] ring-stage parasites were either treated with 2.5 mM GLCN or left untreated and incubated until schizont rupture. Following schizont rupture, Giemsa-stained smears were made of these cultures without washing or removing the culture medium, blinded and counted to determine the number of bound merozoites and newly invaded rings in each treatment. For each treatment >1000 RBCs were counted per replicate.

**Bound merozoite assay and invasion scoring by confocal microscopy**. To determine whether PfCERLI2 knockdown inhibits invasion, prior, at or following tight junction formation, uninfected RBCs were stained in 1 mM CellTrace™ Far Red (Invitrogen) to allow for fluorescent visualisation and downstream 3D modelling of the RBC. Stained RBCs at 6% haematocrit were treated with 1 mM cytochalasin D (ThermoFisher Scientific) or left untreated, then distributed as 75 μL aliquots in a 96-well plate. PfCERLI2[HAGlmS] ring-stages (60 mL) at 4–5%

parasitaemia were treated with 2.5 mM GLCN or left untreated, incubated until young schizonts had formed at the point of chromatin segregation, Percoll-purified, then treated with E64 for 4 h. Viable merozoites[23] were then purified from the E64-treated percoll-purified schizonts by resuspending in 2 mL culture media (excluding Albumax) and passing the cells through a 1.2 mm Minisart® Syringe Filter. 75 μL of flowthrough was immediately added to wells containing CellTrace-stained, cytochalasin D treated or untreated RBCs, now at a final haematocrit of 3%. The plate was shaken at 300 rpm at 37 °C for 2 min to allow for a short invasion window, the cells were then fixed (final concentration 4% w/v paraformaldehyde (PFA, Sigma-Aldrich), 0.0075% v/v glutaraldehyde (Electron Microscopy Sciences), pH 7.4). Slides for confocal microscopy were prepared as described in a following section—stains included anti-AMA1 (mouse 1/500), anti-MSP1-19 (rabbit 1/1000) and DAPI. Slides were blinded shortly after sample preparation, and Z-stacks of 20 merozoites were taken per treatment group on an LSM800 (Zeiss) in Airyscan mode. Invasion was scored under three tiers: 1-attached, 2-partially invaded, 3-invaded, which could be determined by whether the merozoite appeared on the surface of, partway through the interface of, or enveloped inside the CellTrace fluorescently labelled RBC membrane respectively. The diameter of AMA1 of each merozoite was measured in Imaris (Bitplane). Samples were unblinded after statistical analyses were performed. Measurements for merozoites with an invasion score = 3 were excluded from the final data analysis as the tight junction closes behind the merozoite upon completion of invasion and becomes smaller. The full dataset is provided in the Supplementary Material.

**Saponin lysis and western blot**. To prepare protein samples, high parasitaemia cultures were lysed in 0.15% w/v saponin (ThermoFisher Scientific) on ice for 10 min, centrifuged at 16,000 rcf, washed once in 0.075% w/v saponin and twice in PBS. All lysis reagents included protease inhibitors (CØmplete, Roche). Parasite lysates were DNase I treated (Qiagen) for 5 min at room temperature before being resuspended in reducing sample buffer (0.125 M Tris-HCl pH 7, 20% v/v glycerol, 4% v/v SDS, 10% v/v b-mercaptoethanol (Sigma-Aldrich), 0.002% w/v bromophenol blue (Sigma-Aldrich)). Protein lysates were separated by size on SDS-PAGE 4-12% Bis-tris Gels (Bolt, Invitrogen) at 110 V for 80 min then transferred onto a nitrocellulose membrane (iBlot, Invitrogen) at 20 V for 7 min. Membranes were blocked in 1% w/v skim milk 0.05% v/v Tween20 (Sigma-Aldrich) in PBS (hereby referred to as 1% milk PBS-T) for 1 h at room temperature. Primary (mouse 12CA5 anti-HA (1:5000 Roche), rabbit anti-aldolase (1:5000, Abcam), rabbit anti-GAP45[57] (1:5000), mouse anti-RAP1[58] (1:10,000), rabbit anti-RON4[59] (1:5000), rabbit anti-Rh4[60] (1:5000), rabbit anti-EBA175[61] (1:10,000), rabbit anti-MSP1-19 (1:10,000), rabbit anti-EXP2 (1:10,000), rabbit anti-ERC (1:10,000), rabbit anti-GAPDH (1:5000), and secondary (IRDye ® 800CW goat anti-mouse (1:4000, LI-COR Biosciences), IRDye® 680RD goat anti-rabbit (1:4000, LI-COR Biosciences) antibodies were prepared in 1% milk PBS-T and incubated on the membranes for 1 h each at room temperature while rocking. Primary antibodies were washed three times in 0.05% v/v PBS-Tween20 (hereby referred to as PBS-T), while secondary antibodies were washed twice in PBS-T and once in PBS. Completed blots were dried on filter paper (Whatman) and visualised on the Odyssey Infrared Imaging System (LI-COR Biosciences). All band quantification was performed in Image Studio Lite 2.5 (LI-COR Biosciences).

**Proteinase K protection assay**. To determine whether PfCERLI2 was cytosolically exposed or compartmentalised in an organelle, a proteinase K protection assay modified from a previous study[10,13] was performed[10,13]. Three 10 mL aliquots of high schizontaemia cultures were centrifuged at 440 rcf for 5 min, supernatant removed and lysed in 0.15% w/v saponin as described above. One set of schizonts were treated in SOTE (0.6 M sorbitol, 20 mM Tris HCl pH 7.5, 2 mM EDTA) alone, the second treated in SOTE with 0.02% w/v digitonin (Sigma-Aldrich) left incubating for 10 min at 4 °C before being washed in SOTE, and the third digested with 0.1 mg/mL Proteinase K (Sigma-Aldrich) in SOTE for 30 min at 4 °C following the aforementioned digitonin treatment. Proteinase K was inactivated with 50 mL 100% v/v trichloroacetic acid followed by PBS. All samples were resuspended in 500 mL acetone to ensure full Proteinase K deactivation, centrifuged then washed twice in 500 mL MilliQ $H_2O$. The final pellets were used for western blot analysis of PfCERLI2's sensitivity to Proteinase K.

**Protein solubility assay**. To biochemically determine whether PfCERLI2 was membrane-associated and by what mechanism of association, a protein solubility assay was performed as previously described[13]. Briefly, a high schizontaemia culture was saponin-lysed as described above, resuspended in 100 mL MilliQ $H_2O$, snap-frozen in dry ice four times, passed through a 29 gauge needle five times to disrupt parasite membranes, centrifuged at 16,000 rcf for 10 min before reserving the water-soluble fraction containing cytosolic proteins. The remaining pellet was washed twice in MilliQ $H_2O$ and once in PBS before being resuspended in 100 mL 0.1 M $Na_2CO_3$ for 30 min at 4 °C, centrifuged before reserving the carbonate-soluble fraction containing peripherally-associated proteins. The pellet was washed in the same manner above before being resuspended in 0.1% v/v Triton X-100 for 30 min at 4 °C, centrifuged and the final supernatant reserved containing integral proteins. The remaining pellet was washed twice in PBS and was used to represent

Triton X-100-insoluble proteins. All fractions were analysed by western blot to determine PfCERLI2's membrane solubility profile.

**Sample preparation for fixed-cell immunofluorescence microscopy.** *Pfcerli2*-[HAGlmS] cultures of 3% E64 arrested schizonts were centrifuged at 440 rcf for 3 min, washed in PBS, resuspended in fixative (4% v/v paraformaldehyde (PFA, Sigma-Aldrich), 0.0075% v/v glutaraldehyde (Electron Microscopy Sciences), pH 7.4) and then left gently rocking for 30 min at room temperature. Samples were centrifuged at 440 rcf, fixative removed, washed twice in PBS, then adjusted to 1% haematocrit in PBS. #1.5H high-precision coverslips (Carl Zeiss, Oberkochen, Germany) were soaked in methanol, air-dried, coated in 0.01% v/v poly-L-lysine (Sigma-Aldrich) for 30 min at room temperature then washed in MilliQ H$_2$O. Fixed cells were then laid on top at room temperature for 30 min, with non-adherent cells being gently aspirated off. Cells were permeabilised with 0.1% v/v Triton X-100 for 10 min then incubated in 3% bovine serum albumin (BSA) in PBS-T for 1 h. Primary antibodies (anti-HA biotin conjugate 1:1000 (Roche), mouse anti-RAP1[58] 1:500, mouse anti-AMA1[62] 1:500, rabbit anti-RON4[59] 1:500, rabbit anti-MSP1-19[63] 1:500) were diluted in 1% w/v BSA in PBS-T as specified above and applied to the coverslips overnight at 4 °C. The next day, coverslips were washed three times in PBS-T, before being incubated with Alexa Fluor-conjugated secondary antibodies (strep-tavidin 488 nm, mouse 594 nm, rabbit 647 nm, Life Technologies) for 1 h in the dark at room temperature. Coverslips were again washed three times in PBS-T before being dehydrated in ethanol (70% v/v 3 min, 90% v/v 3 min, 100 v/v 3 min), air-dried and then mounted on glass slides with 20 mL Prolong® Gold antifade solution (refractive index 1.4) with 4',6-diamidino-2-phenylindole dihydrochloride (DAPI, ThermoFisher Scientific). The mountant was allowed to cure overnight and coverslips were analysed on an Olympus FV3000 confocal microscope.

**Confocal microscopy and colocalisation analysis.** Confocal microscopy was performed using an Olympus FV3000 fluorescence microscope (Olympus) equipped with a ×100 MPLAPON oil objective (NA 1.4) using the 405, 488, 561 and 633 nm lasers. Colocalisation analysis was performed using Imaris Coloc Suite (v9.0, Bitplane Inc., Switzerland) as described in detail previously[13]. Briefly, images were exported from the Olympus FV3000 in .oir format, before conversion into .ims format. Following conversion, colocalisation analysis was performed whereby the image dataset was masked to the thresholded PfCERLI2 (anti-HA, 488 nm laser) signal, defining the PfCERLI2 signal as the region of interest. Subsequently the other thresholded channels, corresponding to either RAP1, RON4, AMA1 or MSP1-19, were colocalised with the PfCERLI2 signal. Colocalisation statistics reported in this study are Pearson's correlation coefficient between the PfCERLI2 signal and the signal of the corresponding organelle marker, when PfCERLI2 is defined as the region of interest. Colocalisation analyses were performed over three biological replicates. For each of these biological replicates, threshold values were determined for each of the channels used in the colocalisation analysis to differentiate between true signal and background. Threshold values were fixed across all channels for each image in that biological replicate.

**Rhoptry and microneme ligand secretion assay.** Secretion assays were modified from a previous protocol[13]. Briefly, *Pfcerli2*[HAGlmS] synchronous ring stages at 15% haematocrit were grown in 6-well plates (Corning) with or without 2.5 mM GLCN for 24 h. Once trophozoite stages had formed, cultures were centrifuged at 440 rcf, supernatant removed and resuspended in an enzyme mix (0.067 U/mL neur-aminidase (Sigma-Aldrich), 1 mg/mL chymotrypsin (Worthington Biochemical Corporation), 1 mg/mL trypsin (Sigma-Aldrich)) to be incubated at 37 °C for 45 min to cleave RBC surface receptors. Parasites were then washed twice in culture media and returned to culture for a further 24 h until schizont rupture. Once ruptured, cultures were centrifuged at 16,000 rcf for 10 min at 4 °C to pellet uninvaded merozoites. Supernatants containing secreted protein was collected and kept on ice, and the pellet subject to saponin lysis to represent non-secreted protein. Each fraction was analysed by western blot and quantified in Image Studio Lite 2.5 (LI-COR Biosciences).

**Transmission electron microscopy.** *Pfcerli2*[HAGlmS] synchronous ring-stage cultures were treated with or without 2.5 mM GLCN and grown until early schizonts had formed before being treated with 2 µM C1 for 4 h. Late schizonts were then enriched using Percoll then fixed in 2.5% v/v glutaraldehyde (Electron Microscopy Sciences) in PBS overnight at 4 °C. Fixed schizonts were post-fixed in 1% osmium tetroxide (v/v) and 1.5% potassium ferrocyanide (w/v) in 0.1 M Sorensen's phosphate buffer (PB) pH 7.4 for 30 min at room temperature. Cells were washed in 0.1 M PB (3 × 5 min) and incubated in 1% tannic acid (w/v) in 0.1 M PB for 30 min. Samples were washed in ultrapure water (3 × 5 min) and en bloc stained in 1% aqueous uranyl acetate (w/v) for 1 h at room temperature. Cells were washed in ultrapure water (3 × 5 min) and incubated in Walton's Lead Aspartate for 30 min at 60 °C. Cells were then washed in ultrapure water (3 × 5 min), dehydrated in increasing concentrations of ethanol-H$_2$O and absolute acetone and finally embedded in medium-grade Procure 812 resin. 70 nm ultrathin sections were generated using a Leica EM UC7 ultramicrotome (Leica Microsystems) and attached to 100-mesh Cu formvar-carbon coated grids. Images were acquired using a Tecnai G2 F30 transmission electron microscope (Field Electron and Ion

Company, FEI) operating at 200 kV. Rhoptry length was quantified from trans-mission electron micrographs for rhoptries where necks could be seen contacting the merozoite plasma membrane and the bottom of the bulb could be clearly discerned. Using blinded assessment, a vector was drawn from the tip of the necks to the end of the rhoptry bulbs in ImageJ to provide a length in nm. Micrographs where the rhoptries had been cut sideways such that it removed part of the neck or bulb were excluded from this analysis.

**Array tomography and rhoptry morphometric analysis.** Array tomography[64,65] was performed in order to obtain 2-D and 3-D morphometric data. Blocks processed for TEM were sectioned on a Leica EM UC7 ultramicrotome (Leica Microsystems), generating 100 consecutive sections of 100-nm thickness (depth sectioned = 10 µm/sample). Sections were collected on a hydrophilized silicon wafer and allowed to dry at room temperature. An aluminium stub was adhered to the underside of wafers using conductive double-sided adhesive carbon-tabs.

Inverted backscattered field emission scanning electron microscopy was performed using a FEI Teneo SEM (Field Electron and Ion Company, FEI) operating at 5 kV at a working distance of 7.3 mm. Images (6144 × 4096 pixel array, 8-bit, 9.76 × 9.76 × 100 nm voxel dimensions, pixel dwell time 2 µs) were acquired at 3500× yielding a XY field of view of 59.97 × 39.98 µm).

Image histogram stack normalisation was performed, and images were aligned using the StackReg plugin for Fiji[66]. For 3-D modelling and visualisation, datasets were processed using 3dmod, a program bundled with the IMOD software package[65]. Rhoptries were segmented by means of manual tracing of high-contrast lines. 3-D information including surface area and volume was obtained using the get-info command. Rhoptry length was quantified from SEM micrographs where rhoptry necks could be seen contacting the merozoite plasma membrane and the bottom of the bulb could be clearly discerned. Using blinded assessment, a vector was drawn from the tip of the necks to the end of the rhoptry bulbs in 3dmod to provide a length in nm.

**Airyscan microscopy, merozoite measurement and foci analysis.** Airyscan microscopy was performed using a Zeiss LSM800 AxioObserver Z1 microscope (Carl Zeiss, Oberkochen, Germany) that had an Airyscan detector and a Plan-Apochromat 63x M27 oil objective (NA 1.4). Z-stacks were acquired using a 0.04 µm XY and 0.16 µm Z pixel resolution. Images were acquired sequentially as follows: Channel 1 (RON4)—633 nm laser, channel 2 (RAP1)—594 nm laser, channel 3 (HA)—488 nm laser, channel 4 (DAPI)—405 nm laser.

The distance between the basal end of the nucleus (DAPI signal) and the RAP1 or RON4 foci was measured as described previously[67]. Images were first blinded and then using the Plot Profile plugin on FIJI, a line was drawn from the basal extremity of the DAPI signal through the RAP1 and RON4 foci to generate intensity profiles for the three markers as a function of distance of the line. Data generated from these intensity plots were then exported, and graphed using GraphPad PRISM 7, with the distance corresponding to the highest RAP1 or RON4 signal intensity representing the distance of that signal from the basal end of the DAPI signal.

RON4 and RAP1 foci were analysed using an object analysis pipeline on Imaris (v9.8.0, BitPlane AG, Zurich, Switzerland) that had previously been established for these markers[13]. RAP1 or RON4 signals were pre-processed to segment and split individual foci, and then thresholded to remove background signal. Threshold values for each marker were kept consistent across all replicates and treatments. Identified objects were then filtered on the basis of size, with only objects >0.1 µm included in the analysis. This analysis was performed in batch mode, so all images were processed and quantified identically. Once the surfaces had been established, their surface area, volume, and sphericity values were exported as.csv files.

RAP1 diameter was measured using the line function on ImageJ. Briefly, images of PfCERLI2[HAGlmS] parasites that had either been treated with 2.5 mM GLCN or left untreated and stained with anti-RAP1 antibodies were blinded. Rhoptry bulb diameters were then manually quantified by drawing a vector across the diameter of the RAP1 signal. Only rhoptries with visible 'donut-shapes' were measured, with the diameter measured on the Z-slice between the points with the greatest signal intensity. Merozoites that were densely packed inside the schizont, or RAP1 signals where it was unclear which merozoite they belong to, were excluded from the analysis.

**Preparation of samples for DiQ-BioID.** Early PfCERLI1[DiQ-BioID] PfCERLI2[DiQ-BioID] schizonts (36-h old) were treated with 150 µM biotin (Sigma-Aldrich) and with or without rapalog A/C heterodimerizer (AP21967, Takara) for 8 h total, where E64 was added at 4 h prior to treatment. Once mature schizonts formed, dimerisation between the bait protein and BirA was confirmed by live-cell microscopy, and biotinylation confirmed by fixed-cell IFA and western blot using a streptavidin-conjugated AlexaFluor488 and streptavidin-conjugated IRDye800 respectively. Expression of the BirA-FRB-mCherry complex was confirmed by western blot probing with rabbit anti-mCherry (1:5000, BioVision). The remaining culture was centrifuged at 550 rcf, washed twice in PBS before being saponin-lysed.

Saponin pellets were further lysed in 1.5 mL lysis buffer (50 mM Tris-HCl pH 7.5, 500 nM NaCl, 1% Triton X-100 (Sigma-Aldrich), 1 mM dithiothreitol (DTT, Sigma-Aldrich), 1 mM phenylmethylsulfonyl fluoride (PMSF, Sigma-Aldrich) with CØmplete EDTA-free protease inhibitor cocktail tablet (Roche)) and bath sonicated in a Digital Benchtop Ultrasonic Cleaner (Soniclean) three times for 30 s at a time.

Samples were centrifuged at max speed, and supernatants were incubated with magnetic streptavidin beads (Pierce) at 4 °C O/N to enrich for biotinylated proteins. The next day, the beads were separated on a magnetic separation rack, and flowthrough was discarded. Beads were washed three times in 1 mL lysis buffer, and six times in 50 mM ammonium bicarbonate (Sigma-Aldrich). Washed beads were then resuspended in 200 μL ammonium bicarbonate containing 1 μg equivalent of sequencing-grade trypsin (Promega) and incubated O/N at 37 °C with gentle shaking. The following morning, an additional 0.5 μg of sequencing-grade trypsin was added to each sample, and incubated for another 4 h at 37 °C. Beads were magnetised, and supernatant containing peptides was collected. Residual peptides on the beads were collected with two rinses of 150 μL ddH$_2$O and supernatants were pooled. The pooled supernatant was centrifuged at max speed to remove any residual beads and collected in a new tube. The final supernatant was acidified to 0.5% v/v formic acid (Thermofisher) to inactivate trypsin, before being sent for LC-MS analysis.

**Mass spectrometric instrumentation and data acquisition.** Mass spectrometry acquisition and NanoLC-MS/MS was carried out as described previously[68,69], with minor modifications. Samples were loaded at a flow rate of 15 μl/min onto a reversed-phase trap column (100 μm × 2 cm), Acclaim PepMap media (Dionex) and maintained at a temperature of 40 °C. Peptides were eluted from the trap column at a flow rate of 0.25 μl/min through a reversed-phase capillary column (75 μm × 50 cm) (LC Packings, Dionex). For acquisition by HPLC, a 158 min gradient was set using an incremental gradient that reached 30% ACN after 123 min, 34% ACN after 126 min, 79.2% ACN after 131 min and 2% ACN after 138 min for a further 20 min. The mass spectrometer was operated in a data-independent mode where a 43-fixed-window setup of 14 $m/z$ effective precursor isolation over the $m/z$ range of 376–967 Da was applied.

For analysis, raw files were processed using Spectronaut™ (version 13.0) against the in-house generated *P. falciparum* spectral library containing 44,449 peptides corresponding to 4,730 proteins (parasite and human). For processing, raw files were loaded in Spectronaut, the ideal mass tolerances for data extraction and scoring were calculated on its extensive mass calibration with a correction factor of 1. Both at precursor and fragment level, the highest data-point within the selected $m/z$ tolerance was chosen. Identification of peptides against the library was based on default Spectronaut settings (Manual for Spectronaut 13.0, available on Biognosis website). Briefly, precursor Qvalue Cut-off and Protein Qvalue Cut-off were as per default at 1% and therefore only those that passed this cut-off were considered as identified and used for subsequent processing. Retention time (RT) prediction type was set to dynamic indexed RT. Interference correction was performed at the MS2 level. For quantification, interference correction was activated and cross run normalisation was performed using the total peak area at a significance threshold of 0.01. Statistical analysis was performed in excel using relative protein and student $t$-test as described previously[69].

**Statistics and reproducibility.** All graphs and statistical analyses were completed in Graphpad PRISM 7 (GraphPad Software Inc.). All figure legends note the type of statistical test, the definition of significance for various $p$-values, and the number of biological replicates ($n$) for each experiment.

**Reporting summary.** Further information on research design is available in the Nature Research Reporting Summary linked to this article.

## Data availability

The mass spectrometry proteomics data have been deposited to the ProteomeXchange Consortium via the PRIDE[70] partner repository with the dataset identifier PXD028937. Uncropped and unedited blots/gels are included in Supplementary Figs. 18–22. Source data underlying the main Figures are available in Supplementary Data File 5. All other data available on request.

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

## Acknowledgements

We thank Prof. Alan Cowman for provision of RON4, EBA175 and GAP45 antibodies, A/Prof. Wai-Hong Tham for RH4 antibodies, Dr. Matt Dixon for GAPDH and ERC antibodies. We also thank Dr. Paul Gilson for the PTEX150HAGlmS transfection vector, MSP1-19 and EXP2 antibodies. Fluorescence imaging was performed at Adelaide Microscopy (The University of Adelaide) and the Centre for Cancer Biology (UniSA). Electron microscopy was performed at the Ian Holmes Imaging Centre, Bio21, The University of Melbourne (www.microscopy.unimelb.edu.au). For provision of the SLI-TGD vector, we thank Dr. Tobias Spielmann. We thank Dr. Brad Sleebs for Compound 1. We thank the Australian Red Cross Blood Bank for the provision of human blood that was collected with informed consent of the donors. We thank the VEuPathDB team, as access to their database enabled much of this research. This work was supported by funding from the NHMRC (Project Grant APP1143974, D.W.W.), University of Adelaide Beacon Fellowship and Hospital Research Foundation Fellowship (D.W.W.), DAAD/Universities Australia joint research co-operation scheme (T.G., D.W.W., B.L., J.B.), Australian Government Research Training Program Scholarship (B.L., J.B.), South Australian Commonwealth Scholarship (B.L.), DFG BA5213/3-1 (J.S.W.) and Jürgen Manchot-Stiftung fellowship (A.A.).

## Author contributions

Study design and planning: D.W.W., B.L., J.B., G.S., D.J.C., M.W.A.D., L.T. and T.G. Performed experiments and generated reagents: B.L., J.B., D.W.W., G.J.S., J.S., G.S., E.E., G.K.H., A.A. and J.S.W. Data analysis: B.L., J.B., G.J.S., G.S., J.S., S.F. and D.W.W. Manuscript writing: B.L., J.B., G.S., T.G. and D.W.W. Manuscript was drafted with input from all authors.

## Competing interests

The authors declare no competing interests.
