## [Transparent Peer Review File · Communications Biology]

Reviewers' comments:

Reviewer #1 (Remarks to the Author):

This study by Liffner et al, focuses on a rhoptry bulb protein termed PfCERLI2. By using sequence data, phylogenetic analyses and structure predictions they suggest that this protein is a paralogue of PfCERLI1, which has recently been characterised (Suarez et al, 2019 and Liffner et al, 2020). Utilising conditional genetics, the authors show that this protein is essential and involved in merozoite invasion. They continue the characterisation of PfCERLI2 by super-resolution microscopy, EM and biochemical techniques to gain insight on the role of this protein. They demonstrate that it is localised to the rhoptry bulb and is exposed to the merozoite cytosol. Knockdown of PfCERLI2 leads to an altered rhoptry morphology and also inhibits processing of another rhoptry bulb protein, RAP1. The manuscript contains an impressive amount of data regarding PfCERLI2 and its role in merozoite invasion. Having said that, I am a bit confused on the actual biology/role of this protein and certain aspects need clarification to strengthen the conclusions of the authors.

Major points:

1. Looking closely at the phenotypes of PfCERLI1 and PfCERLI2 knockdown as described by the authors in this publication and in the paper by Liffner et al, one can assume that they are almost identical. Both inhibit merozoite invasion (at which stage it is currently unknown for both mutants) with similar % inhibition. They are localised to the rhoptry bulb, have close localisation to RAP1 and are peripherally associated with the cytosolic face of the rhoptry bulb membrane. Both inhibit RAP1 processing and alter rhoptry antigen distribution. The only notable difference is that inhibition of PfCERLI1 affects secretion of rhoptry antigens (but see point 3) and PfCERLI2 inhibition leads to longer and narrower rhotries. For 2 proteins that the only common feature is a C2 domain and a PHIS sequence (the function of which is unknown) this functional overlap is truly amazing. Can the authors provide any evidence that PfCERLI1 is functional in the PfCERLI2 knockdown? Have the authors performed any pull-downs to see if PfCERLI1/2/RAP1 form a complex? Is it possible that this is a double knockdown evidently and that the two proteins do not have the exact same function but act as a complex?
2. Can the authors rescue the phenotype by either overexpressing PfCERLI1 or 2? This can be performed also episomally with a strong promoter (AMA1 for example). It will be interesting to see if overexpression of PfCERLI1 can rescue or reduce the effect seen on the PfCERLI2 knockdown.
3. In Fig. 6b the authors quantify the Western blots from 6a based on the aldolase control. Why is there this variation in the pellet samples (for example from ~20% to 150% for RH4)? It is not clear to me from the methods or from the results why this variation is expected. Can the authors perform any statistical analysis to see if the results are significant or not? Have the authors tried to normalise with another marker such as ERC, in order to compare the results with PfCERLI1?
4. The authors by using Giemsa stained smears suggest that "knockdown inhibits invasion following tight junction formation" (line 295). Yet, in lines 297-299 they conclude that PfCERLI2 knockdown inhibits merozoite invasion "both at and prior to formation of the tight-junction". Giemsa stains cannot provide any evidence regarding the temporal events leading to invasion. You cannot extrapolate from Giemsa stains if these free merozoites are actually attached to RBCs and cannot re-orientate or are just free merozoites close to an RBC. The authors should perform invasion videos or at least IFAs of invading merozoites to show at which point invasion is inhibited.

Minor points:

1. Lines 56-60: A reference will be beneficial here for the non-expert reader.
2. Line 368: MSP1 does not have a transmembrane domain and text should be modified accordingly to "GPI-anchored". Can the authors speculate on why is MSP1 detected in all fractions?

3. Fig 4a. In most IFAs of early schizonts, the PfCERLI2 signal is only visible in few merozoites. Is this a technical issue, since the super resolution images show expression in all merozoites? Can the authors provide any better IFAs? On the AMA1 panel it seems that in the merged image AMA1 is shown as green instead of magenta. If the case, this should be corrected. On a similar note, AMA1 might not be the best micronemal marker as it is expressed quite late in the life cycle. Although not essential to this study, are there any other micronemal antibodies available that authors could use?
4. Fig. 7i&j: Representative images as a supplement should be provided.
5. The loss of HA signal in late schizonts while still detectable in WB is intriguing. The authors speculate masking of the epitope. For this to occur it would suggest formation of a complex leading to the HA not being accessible. Do the authors have another possible explanation that they could speculate on? Relevant to this, in Figure 6a the HA western blot seems to suggest there is a doublet, but in other Western blots this doublet is not present. Is the doublet an artefact? Do the authors have any evidence for proteolytic processing of PfCERLI2?
6. The analysis on the repeats (lines 180-195) is not integral to the paper and should be compressed focusing on the ratio of mutations and the key points as no function can be attributed to them.
7. Supplementary Fig. 5. From the predicted modelling of the whole PfCERLI2 structure and the PH domain structure, I would be very cautious on describing the PH domain as degenerate and non-functioning. It seems as if is completely absent. If the authors feel that it is a degenerate domain they should provide a superimposed image of both to show similarities in the fold. They should also highlight the N and C terminus in the models and the figure legend.
8. The inhibition of RAP1 processing is intriguing. Are both processing events inhibited? Can the authors check if Plasmepsin IX is still in the rhoptries or can they speculate on this part of the phenotype?

Reviewer #2 (Remarks to the Author):

Summary

The study by Liffner et al. describes the characterization of the malaria protein, PfCERLI2. The authors outline their phylogenetic analyses to show that cerli2 likely resulted from a duplication event of cerli1. The authors further create a knockdown parasite line, after unsuccessful attempts at knocking out the gene (indicating essentiality to the parasite) with which they were able to conduct a series of studies to investigate the localization and function of PfCERLI2. These studies found PfCERLI2 to be localized to the cytosolic side of the rhoptry bulb, essential for invasion and deficiency alters rhoptry morphology.

Overall, the manuscript is well-written and there is a logical flow between sections. I believe the work to be very interesting and that it adds another phenotype to the ever-growing body of basic malaria biology literature. The phylogenetic analysis was an insightful addition, although I'm a little baffled by why the parasite would duplicate a gene and have them both be essential. I'm most interested in seeing what the significance of the decapeptide repeat region is towards PfCERLI2 function, as well as whether PfCERLI2 and PfCERLI1 complex together, similarly to their orthologues TgRASP1/RASP2.

Specific comments:

1. I know that the authors are speaking generally, but the last half of the second paragraph of the Introduction could still use some references. Further, the use of a semicolon (;) in line 55 is unnecessary and would be better replaced with a comma (,).
2. Line 56 - "...are dual club-shaped" sounds incomplete.
3. Malaria phylogenetics isn't something I'm particularly well versed at, so I wasn't aware that Laverania was a subgenus of Plasmodium. I think it'd be helpful if there was a brief sentence about it for readers like myself.
4. Line 118 "...sequence identities werelow..."
5. Is the growth of PfCERLI2HAGlMS with glucosamine delayed compared to no glucosamine? Also, do

the treated parasites morphologically appear normal throughout the life cycle prior to invasion?

6. The authors state that expression of PfCERLI2 was reduced by ~72%, based upon the western blot. How exactly was this number derived? I'm assuming that bands were normalized to the loading control (ERC in this case) and quantified, but I didn't see an indication of how many biological replicates. Also, do the authors know if all parasites show the same level of reduction? If a treated parasite still has a little PfCERLI2, is this enough to help it proceed through invasion?

7. I was confused by the addition of the PfCERLI2 growth curve (first pictured in Fig 2F) to Fig 3a. I don't think this is necessary and ideally, I would have preferred to see if WT treated with Glcn showed no differences in invasion.

8. The authors note that the invasion assay had a shorter Glcn treatment, compared to the growth assay (24h vs 72h) and that this may be a reason for reduction being lower than growth inhibition (30% invasion reduction vs ~50% growth inhibition). The western blot (Fig2e) was accomplished by treating ring stage to schizonts (~48h) (line 209), from which expression was said to be reduced by 72%. Could the authors explain why for the invasion assay they did not do a similar treatment (ring-stage to ring stage of the next cycle? What are the expression levels of PfCERLI2 in parasites treated with Glcn from trophozoite to ring-stage (24h treatment)? Further, were successful invasions after Glcn treatment able to progress normally through the next life cycle or does a % not grow properly after invasion?

Sidenote: I feel that this is a disadvantage in using the ribozyme system for exploring protein functions and think that the use of a conditional knock out system (e.g., DiCre) would have been better in getting a clearer idea of the effects of deficiency.

9. Please clarify the treatment time for the merozoite quantity experiment.

10. Invasion is affected as evidenced by the decrease in number of successful invasion events, however it is unclear to me at which point invasion is affected. Have the authors considered live imaging to measure the duration of pre-invasion between treated/untreated parasites? Do the merozoites bind initially to RBCs and then fall off? Or do the parasites remain bound to the RBC membrane?

11. In Fig 5a (proteinase protection assay), the authors use ERC as a control to show that organelle contents are unaffected. Have the authors also tested a rhoptry protein (e.g., RAP1) to show that the rhoptry contents itself are not affected by proteinase treatment? I would think that this be the more appropriate control given that you're asserting that PfCERLI2 is outside, not inside, the rhoptries. Additionally, there appears to be less ERC after proteinase treatment, compared to no treatment. Could the authors comment on this?

12. The authors have used flow cytometry in a number of these studies; however, I have not seen any representative plots to show their gating strategies.

Response to Reviewer's Comments

We thank the Reviewers and Editor for their comments and feedback. We provide a detailed and itemised description of the changes made to the manuscript to address the feedback. All changes can be cross-referenced following the line numbers listed in the rebuttal through highlighting of the changes in the manuscript and Supplementary materials.

We carefully edited the manuscript according to their suggestions and included the following additional data to address the major concerns:

- Proximity Ligation analysis of PfCERLI1 and PfCERLI2 interacting partners to investigate whether the two proteins interact directly.
- Super-resolution imaging of invading merozoites with, and without, PfCERLI2 knockdown to address whether the tight junction can form with PfCERLI2 knockdown.

Additionally, we have clarified interpretation of the bound merozoite data, interpretation of the PH domain loss in PfCERLI2 and included RAP1 as a proteinase protection assay control in Figure 5, as well addressing all reviewers comments.

Reviewer #1 major comments:

1. **Looking closely at the phenotypes of PfCERLI1 and PfCERLI2 knockdown as described by the authors in this publication and in the paper by Liffner et al, one can assume that they are almost identical. Both inhibit merozoite invasion (at which stage it is currently unknown for both mutants) with similar % inhibition. They are localised to the rhoptry bulb, have close localisation to RAP1 and are peripherally associated with the cytosolic face of the rhoptry bulb membrane. Both inhibit RAP1 processing and alter rhoptry antigen distribution. The only notable difference is that inhibition of PfCERLI1 affects secretion of rhoptry antigens (but see point 3) and PfCERLI2 inhibition leads to longer and narrower rhoptries. For 2 proteins that the only common feature is a C2 domain and a PHIS sequence (the function of which is unknown) this functional overlap is truly amazing.**

Response: We agree with the reviewer that, although we provide a number of lines of evidence that PfCERLI2 arose from a gene-duplication of PfCERLI1 and they are functionally related, the degree of functional relatedness is still of great interest given the sequence differences between the proteins. We show that knockdown of both of these proteins using the glucosamine inducible riboswitch leads to a similar, but not identical, knockdown phenotype. There could be a number of reasons for this including different expression levels of the two proteins (there is clearly a lot less CERLI2), exact timing of expression, differences in knockdown efficiency between the two CERLIs and differences in protein function. Because of this, while there are strong similarities in the phenotypes, we are cautious about implying they are identical given these variables. From this comparative analysis, we think the most likely explanation is that CERLI2 has undergone sub-functionalisation from CERLI1, such that these proteins likely have overlapping but not identical functions. The following addresses the reviewers questions more directly:

Have the authors performed any pulldowns to see if PfCERLI1/2/RAP1 form a complex?

Response: To address this important point we undertook proximity ligation assay (PLA) BioID experiments using PfCERLI1 and 2 as bait. The results of these BioID experiments have been delayed due to repeat Sars-COV2 lock-downs in Melbourne where our proteomics collaborators at Monash University work and a lock-down in Adelaide, with these disruptions resulting in minimal access to tissue culture and proteomics facilities.

Both proteomic data sets identified >20 high confidence (>2 peptides with > 10% protein coverage represented) interacting candidates, including ROP14 as a mutual interactor. Interestingly, PfCERLI1 BioID did not identify PfCERLI2 as an interacting partner. Likewise PfCERLI2 BioID did not identify PfCERLI1 as an interacting partner, arguing against a direct CERLI1/CERLI2 interaction that was reported for the *Toxoplasma gondii* orthologues using HA-tag immunoprecipitation in solubilised cellular material (Suarez *et al.* 2019). This difference might be explained by differences in rhoptry biology of these two apicomplexan parasites. There could also be other explanations such as: 1) PfCERLI2 expression is low which may complicate detection, 2) any interactions between PfCERLI1 and PfCERLI2 may be beyond the radius of activity of the BirA biotinylator. For instance, a complex between PfCERLI1 and 2 may involve other proteins which could spatially separate the proteins such that PLA-BioID does not identify the presence of both proteins in the complex. 3), since PLA-BioID was undertaken with intact cells, the native distribution of PfCERLI1 and PfCERLI2 may result in minimal interaction compared to what happens in a complex mixture of solubilised cellular material used in IP.

To reflect this new data we have added the following text:

Results:

Line 567: “Proximity dependent biotin interaction studies fail to show an interaction

between PfCERLI1 and PfCERLI2

*Given that both PfCERLI1 and PfCERLI2 share a localisation on the cytosolic face of the rhoptry bulb membrane and exhibit several similar knockdown phenotypes, we sought to determine whether the two proteins interacted as a complex in this niche. To assay for an interaction, we utilised dimerization induced quantitative proximity-dependent biotin identification (DiQ-BioID), a recently developed method in the field (Figure 8a) ²⁹, and generated transgenic parasite lines, *Pfcerli1^{DiQ-BioID}* and *Pfcerli2^{DiQ-BioID}*, which each separately express a bait PfCERLI1 and PfCERLI2 tagged with an FK binding protein (FKBP) and GFP. These lines are then co-transfected with an episomally-expressed BirA biotin ligase fused to an FKBP rapamycin-binding protein domain (FRB) and mCherry (Supplementary Figure 15a). Integration or episomal maintenance of these constructs were confirmed by PCR (Supplementary Figure 15b).*

Upon the addition of rapalog, the bait and biotinylator are able to dimerise, and the system can then be validated by live fluorescence of colocalising GFP/mCherry foci (Figure 8b). BirA, in the presence of excess biotin, then proceeds to biotinylate proteins in close proximity to the FKBP tagged bait protein, which we verified in fixed cell IFA using a streptavidin-

conjugated fluorophore signal colocalising with RAP1 (Figure 8c). To prepare samples for proteomics, 36 hour-old parasites were double-treated with rapalog and biotin for 8 hours, including 4 hours on E64, before being processed for streptavidin-based pulldown and analysis by mass spectrometry.

To minimise the discovery of false positives from non-specific biotinylation, hits were first normalised against samples without rapalog treatment and then to a BirA-control line that has no FKBP tagged bait protein. A representative western blot for the extent of biotinylation between test and control samples are included in Supplementary Figure 15c. In order to identify proteins that were reliably enriched, we only considered hits which were identified with at least 2 peptides with an overall peptide coverage of 10% or more from the in-house *P. falciparum* library for all biological replicates (replicate comparison Supplementary Figure 16). Applying these criteria generated an interactome of 23 proteins for PfCERLI1 (Figure 8d), and 26 proteins for PfCERLI2 (Figure 8e), which included the bait proteins themselves. PfCERLI2 was not identified as an interactor for PfCERLI1 and vice versa. Nevertheless, several rhoptry and schizont-expressed proteins were identified for both data sets to highlight the target specificity of DiQ-BioID. Of greatest interest includes RON11, ROP14 and the cAMP-dependent catalytic subunit of protein kinase A (PKAc) as potential interactors of PfCERLI1. Likewise, PfCERLI2 appears to interact with ROP14. Both lists display several uncharacterised proteins of unknown function that have peak expression at schizogony.”

Figure Legend:

“Figure 8: DiQ-BioID of PfCERLI1 and PfCERLI2 reveals their protein interactome.

(a) A schematic of the DiQ-BioID system under control and test conditions. (b) *Pfcerli1*^{DiQ-BioID} and *Pfcerli2*^{DiQ-BioID} parasites were treated for 8 hours on rapalog to permit for dimerisation between the GFP-tagged bait construct and mCherry-tagged biotinylator, visualised by live fluorescence. Scale bars = 2 μ m. (c) Biotinylation at the rhoptry bulb was confirmed by co-localising foci between RAP1 staining and a streptavidin-conjugated AlexaFluor488 in fixed cells by IFA. Scale bars = 2 μ m. (d) Interactors for PfCERLI1. N = 3 biological replicates. (e) Interactors for PfCERLI2. N = 3 biological replicates. (d-e) Colour code corresponds to: green (bait protein), yellow (peak schizont-expressed), pink (late trophozoite to schizont stage-expressed), blue (uniformly or ring stage-expressed). * P-value calculated by t-test between intensity values in double-treated parasites and no rapalog controls. Proteins not identified in the no rapalog control were assigned with an intensity value of 0.”

Discussion:

Line 716: “... Our DiQ-BioID data for PfCERLI1 and PfCERLI2 currently does not support a direct interaction between these proteins in *P. falciparum*. However, it is still possible that the proteins interact via another protein or a complex that may not be detected by DiQ-BioID in intact cells but can immunoprecipitated in solubilised cells using a HA-tag. Low expression of PfCERLI2 is another possible explanation. For the coccidia and haematozoa containing cerli paralogues, it is likely that the duplicated cerli2 locus that arose from gene duplication of an ancestral cerli1 locus underwent subfunctionalisation. This is supported by shared characteristics of PfCERLI1 and PfCERLI2, along with the strong negative selection CERLI2 has been under across Apicomplexa and the presence of only a putative remnant of a PH domain.

Dimerization-induced proximity based biotinylation using either PfCERLI1 or PfCERLI2 as the bait delivered a list of putative interacting proteins. The most notable one is the multi-transmembrane domain containing ROP14, which appears in the interactomes for both proteins. Though the function of ROP14 remains largely unknown, it has been localised in *P. falciparum* schizonts by IFA and denoted to be slightly posterior to RON4³² and therefore may possess a rhoptry bulb localisation. It is possible that ROP14, in complex with other proteins, forms either a bridge between PfCERLI1 and PfCERLI2 that was not detected within the biotinylation range of BirA, which can vary from 10-30 nm³³. Alternatively, ROP14 may transiently interact with both proteins, but not simultaneously.”

Materials and Methods:

Line 1132: “Preparation of samples for DiQ-BioID

Early PfCERLI1^{DiQ-BioID} PfCERLI2^{DiQ-BioID} schizonts (36 hours old) were treated with 150 μ M biotin (Sigma-Aldrich) and with or without rapalog A/C heterodimerizer (AP21967, Takara) for 8 hours total, where E64 was added at 4 hours into treatment. Once mature schizonts had formed, dimerisation between the bait protein and BirA was confirmed by live cell microscopy, and biotinylation confirmed by fixed cell IFA and western blot using a streptavidin-conjugated AlexaFluor488 and streptavidin-conjugated IRDye800 respectively. The remaining culture was centrifuged at 550 rcf, washed twice in PBS before being saponin lysed.

Saponin pellets were further lysed in 1.5 mL lysis buffer (50 mM Tris-HCl pH 7.5, 500 nM NaCl, 1% Triton X-100 (Sigma-Aldrich), 1 mM dithiothreitol (DTT, Sigma-Aldrich), 1 mM phenylmethylsulfonyl fluoride (PMSF, Sigma-Aldrich) with CØmplete EDTA-free protease inhibitor cocktail tablet (Roche)) and sonicated three times for 30 seconds at a time. Samples were centrifuged at max speed, and supernatants were incubated with magnetic streptavidin beads (Pierce) at 4 °C O/N to enrich for biotinylated proteins. The next day, the beads were separated on a magnetic separation rack, and flowthrough was discarded. Beads were washed three times in 1 mL lysis buffer, and six times in 50 mM ammonium bicarbonate (Sigma-Aldrich). Washed beads were then resuspended in 200 μ L ammonium bicarbonate containing 1 μ g equivalent of sequencing-grade trypsin (Promega) and incubated O/N at 37 °C with gentle shaking. The following morning, an additional 0.5 μ g of sequencing-grade trypsin was added to each sample, and incubated for another 4 hours at 37 °C. Beads were magnetised, and supernatant containing peptides was collected. Residual peptides on the beads were collected with two rinses of 150 μ L ddH₂O and supernatants were pooled. The pooled supernatant was centrifuged at max speed to remove any residual beads and collected in a new tube. The final supernatant was acidified to 0.5% v/v formic acid (ThermoFisher) to inactivate trypsin, before being sent for LC-MS analysis.

Mass spectrometric instrumentation and data acquisition

Mass spectrometry acquisition and NanoLC-MS/MS was carried out as described previously^{64,65}, with minor modifications. Samples were loaded at a flow rate of 15 μ L/min onto a reversed-phase trap column (100 μ m \times 2 cm), Acclaim PepMap media (Dionex) and maintained at a temperature of 40°C. Peptides were eluted from the trap column at a flow rate of 0.25 μ L/min through a reversed-phase capillary column (75 μ m \times 50 cm) (LC Packings,

Dionex). For acquisition by HPLC, a 158 min gradient was set using an incremental gradient that reached 30% ACN after 123 min, 34% ACN after 126 min, 79.2% ACN after 131 min and 2% ACN after 138 min for a further 20 min. The mass spectrometer was operated in a data-independent mode where a 43-fixed-window setup of 14 m/z effective precursor isolation over the m/z range of 376-967 Da was applied.

For analysis, raw files were processed using Spectronaut™ (version 13.0) against the in-house generated *P. falciparum* spectral library containing 44,449 peptides corresponding to 4,730 proteins. For processing, raw files were loaded in Spectronaut, the ideal mass tolerances for data extraction and scoring were calculated on its extensive mass calibration with a correction factor of 1. Both at precursor and fragment level, the highest data-point within the selected m/z tolerance was chosen. Identification of peptides against the library was based on default Spectronaut settings (Manual for Spectronaut 13.0, available on Biognosis website). Briefly, precursor Qvalue Cut-off and Protein Qvalue Cut-off were as per default at 1% and therefore only those that passed this cut-off were considered as identified and used for subsequent processing. Retention time (RT) prediction type was set to dynamic indexed RT. Interference correction was performed at the MS2 level. For quantification, interference correction was activated and cross run normalisation was performed using the total peak area at a significance threshold of 0.01. Statistical analysis was performed in excel using relative protein and student t-test as described previously⁶⁵.”

Reference 29: Birnbaum, J. *et al.* A Kelch13-defined endocytosis pathway mediates artemisinin resistance in malaria parasites. *Science* **367**, 51-59, doi:doi:10.1126/science.aax4735 (2020).

Reference 32: Zuccala, E. S. *et al.* Subcompartmentalisation of Proteins in the Rhoptries Correlates with Ordered Events of Erythrocyte Invasion by the Blood Stage Malaria Parasite. *PLOS ONE* **7**, e46160, doi:10.1371/journal.pone.0046160 (2012).

Reference 65: Birrell, G. W. *et al.* Multi-omic Characterization of the Mode of Action of a Potent New Antimalarial Compound, JPC-3210, Against *Plasmodium falciparum*. *Mol Cell Proteomics* **19**, 308-325, doi:10.1074/mcp.RA119.001797 (2020).

Is it possible that this is a double knockdown evidently and that the two proteins do not have the exact same function but act as a complex?

Response: Although our BiID results indicate that PfCERLI1 and PfCERLI2 do not interact directly, there remains a possibility that PfCERLI1 and PfCERLI2 are part of a broader complex. However, we only really observed one protein biotinylated in both proximity ligation assays, ROP14, which does not provide strong evidence for the existence of a larger, shared, complex that both PfCERLI1 and PfCERLI2 interact with. Therefore, our current interpretation is that the exact functions of these proteins is likely to have diverged, leading to protein interaction that are largely non-overlapping between the two.

2. Can the authors provide any evidence that PfCERLI1 is functional in the PfCERLI2 knockdown?

And

Can the authors rescue the phenotype by either overexpressing PfCERLI1 or 2? This can be performed also episomally with a strong promoter (AMA1 for example). It will be interesting to see if overexpression of PfCERLI1 can rescue or reduce the effect seen on the PfCERLI2 knockdown.

This would indeed very interesting, although we have experienced significant and technical problems that we have not been able to overcome in addressing this suggestion. As suggested, we have tried to perform a phenotype rescue experiment. Despite repeated attempts, trouble shooting and recloning of the vectors, we have been unable to episomally express PfCERLI1 or PfCERLI2 using established vector systems such as the pARL-AMA1 (Geiger *et al.* 2019). This has prevented us from testing whether complementation with PfCERLI1 will rescue the PfCERLI2 knockdown phenotype (and vice versa) or use the lines in to co-localise PfCERLI1 or 2 in PfCERLI2 knock-down parasites. It is important to note here that our BioID results indicate that PfCERLI1 and PfCERLI2 do not interact directly. However, there remains a possibility that both PfCERLI1 and PfCERLI2 are part of a broader complex with functional overlap in this complex. Given the potential scope of investigating this complex, we have maintained the focus on characterising PfCERLI2 function to a high level in this manuscript, with further investigation of the broader function of PfCERLI1, PfCERLI2 and interacting proteins to be covered in future studies.

Geiger, M. *et al.* Structural Insights Into PfARO and Characterization of its Interaction With PfAIP. *Journal of Molecular Biology*, doi:<https://doi.org/10.1016/j.jmb.2019.12.024> (2019).

- 3. In Fig. 6b the authors quantify the Western blots from 6a based on the aldolase control. Why is there this variation in the pellet samples (for example from ~20% to 150% for RH4)? It is not clear to me from the methods or from the results why this variation is expected. Can the authors perform any statistical analysis to see if the results are significant or not? Have the authors tried to normalise with another marker such as ERC, in order to compare the results with PfCERLI1?**

Response: We think that Aldolase is an appropriate control for this experiment since it is present in both the pellet and supernatant fractions, whereas ERC is only present in the pellet fraction (Liffner *et al.*, 2020). The use of Aldolase as a loading control allows normalisation of secreted proteins to the supernatant aldolase, and non-secreted proteins to the pellet aldolase.

The observed variability in these experiments are less than ideal but we would like to communicate that these assays are inherently variable, possibly due to variations in levels of protein knock-down, the pellet sample harvests, loading, antibody binding kinetics and fluorescence development. Similar variability was seen in the equivalent experiments for PfCERLI1 (Liffner *et al.* Nature Communications, 2020). To provide the most reproducible and reliable experimental data possible, we show the combined and individual datapoints for 6 independent experiments. Importantly, we found that protein levels in pellets and secreted protein levels to be similar between PfCERLI2 knock-down and no knock-down parasites

across the 6 experiments, as would be expected if there is little to no effect on secretion with PfCERLI2 knock-down.

Liffner, B. *et al.* PfCERLI1 is a conserved rhoptry associated protein essential for Plasmodium falciparum merozoite invasion of erythrocytes. *Nature Communications* **11**, 1411, doi:10.1038/s41467-020-15127-w (2020).

- 4. The authors by using Giemsa stained smears suggest that “knockdown inhibits invasion following tight junction formation” (line 295). Yet, in lines 297-299 they conclude that PfCERLI2 knockdown inhibits merozoite invasion “both at and prior to formation of the tight-junction”. Giemsa stains cannot provide any evidence regarding the temporal events leading to invasion. You cannot extrapolate from Giemsa stains if these free merozoites are actually attached to RBCs and cannot re-orientate or are just free merozoites close to an RBC. The authors should perform invasion videos or at least IFAs of invading merozoites to show at which point invasion is inhibited.**

Response: We would like to clarify the interpretation of this data. We first measured the increase in free merozoites with CERLI2 knock-down by flow-cytometry. We found that while there was an increase in the number of free merozoites compared to – GLCN control, it was only about half of that expected given the overall level of invasion inhibition. We interpret that the increase in free merozoites is because invasion was inhibited prior to tight junction formation.

To further dissect temporal events leading to invasion, we undertook fixed-cell experiments using purified merozoites. We chose to look at a marker of tight junction formation and invasion progress between – GLCN and + GLCN treatments, the AMA1 ring (Riglar *et al.* *Cell Host & Microbe*, 2011). We fixed purified merozoites 2-minutes post mixing with RBCs (membrane stained with CellTrace) for parallel – GLCN and + GLCN treatments. Treatments were blinded, and then we stained for AMA1 and measured the diameter of the AMA1 ring. We found that – GLCN treatment resulted in a wide spread of AMA1 diameter measurements. This is expected since the AMA1 ring is small at the start (initial tight junction formation) and end (resealing of the RBC membrane at the now closed off tight junction) of invasion, while the AMA1 ring would be widest near the mid-point of invasion as the parasite traverses through the tight junction. In contrast, the spread of AMA1 ring diameter was greatly reduced in cytochalasin D-treated parasites, which inhibits actin motility and invasion progression, but still permits the formation of the tight junction.

The average size and spread of AMA1 ring diameter was lowest for + GLCN and + GLCN/CytoD treatments. At the same time as measuring AMA1 diameter, we also performed a blind scoring of how far into RBC entry the attached merozoite was (attached, partially invaded, invaded (e.g. completely RBC membrane bound)). The majority of – GLCN parasites were invaded (6) and partially invaded (5). The majority of CytoD (12), + GLCN (11) and + GLCN/CytoD (10) were attached, with only two from the three invasion inhibitor treatments found to be invaded (RBC membrane covering the whole merozoite). Since fully invaded merozoites would be expected to have a small AMA1 ring after the tight junction has closed behind them, reducing the average AMA1 diameter of the - GLCN, we removed the AMA1

diameter measurements for fully invaded merozoites from all treatments for further analysis. When the measurements for fully invaded merozoites was removed, the average AMA1 diameter difference was found to be significant between + GLCN and -GLCN treatments, supporting that PfCERLI2 knock-down stops invasion progression for some parasites early after tight junction formation (Fig 3 l, j, k). In the interest of data transparency, we provide the full AMA1 diameter data set in Supplementary figure 9c.

It is important to note that attached parasites in the + GLCN treatment likely represent only a proportion of the parasites that failed to invade with PfCERLI2 knock-down based on flow-cytometry and Giemsa-smear data, with the remainder resulting in an increase in the number of free merozoites in the culture media. Therefore, we would like to maintain the conclusion that PfCERLI2 knock-down leads to loss of invasion both at or before tight junction formation.

We have now incorporated this data into the section following the flow-cytometry and Giemsa experiments as follows:

Gao, X., *et al.* Triggers of key calcium signals during erythrocyte invasion by *Plasmodium falciparum*. *Nature Communications* 4:2862 (2013).

Riglar, D. T., *et al.* Super-Resolution Dissection of Coordinated Events during Malaria Parasite Invasion of the Human Erythrocyte. *Cell Host & Microbe* 9(1): 9-20 (2011).

We have now incorporated this data into the section following the flow-cytometry and Giemsa experiments as follows:

Results:

Line 328: “...Although Giemsa smears demonstrated an increased in attached merozoites that had failed to invade with PfCERLI2 knockdown relative to control, they cannot accurately show whether the tight junction has formed. Therefore, we used a cytochalasin D-based strategy in fixed cells and analysed invading merozoites by Airyscan super-resolution microscopy (Figure 3i-k). Viable GLCN treated and untreated PfCERLI2^{HAGImS} merozoites were purified and permitted to invade RBCs stained with CellTrace™ Far Red (Figure 3i), and treated with or without cytochalasin D, a potent inhibitor of actin polymerisation which prevents completion of invasion²⁰. CellTrace has been reported to not affect invasion and is a suitable stain to visualise the RBC²¹. Invasion scoring was performed and demonstrated that a large relative proportion of GLCN treated parasites were unable to fully invade the RBC (Figure 3j), consistent with earlier results. Following from our Giemsa-based observations, GLCN merozoites appeared to be able to successfully reorientate their apical tip to the RBC surface and thus we speculate that PfCERLI2 knockdown may inhibit invasion downstream of this step, for example at tight junction formation. To quantify this observation, the diameter of AMA1 staining was measured and used as a proxy to determine progression through the tight junction ring – where a larger ring indicates successful progression. However, as the tight junction ring ratchets itself behind the merozoite at the completion of invasion, potentially becoming smaller, we have excluded data points from merozoites with an invasion score = 3 (fully invaded) from all samples for this analysis (full data in Supplementary Figure 9c). Using this analysis, we found that PfCERLI2 knockdown parasites have a smaller AMA1 diameter compared to untreated parasites, results that mimic those of cytochalasin D treated parasites which blocks entry after formation of the tight junction (Figure 3k). Put together, these data indicate that GLCN inducible PfCERLI2 knockdown inhibits invasion at, or prior to, the formation of the tight junction.”

Reference 20: Boyle, M. J. *et al.* Isolation of viable *Plasmodium falciparum* merozoites to define erythrocyte invasion events and advance vaccine and drug development. *Proceedings of the National Academy of Sciences* **107**, 14378-14383, doi:10.1073/pnas.1009198107 (2010).

Reference 21: Thiam, L. G. *et al.* Cell trace far-red is a suitable erythrocyte dye for multi-color *Plasmodium falciparum* invasion phenotyping assays. *Experimental Biology and Medicine* **245**, 11-20, doi:10.1177/1535370219897393 (2020).

Our strategy for our revised experiment has been added the methods and now reads:

Materials and Methods:

Line 915: ***Bound merozoite assay and invasion scoring by confocal microscopy***

To determine whether PfCERLI2 knockdown inhibits invasion, prior, at or following tight junction formation, uninfected RBCs were stained in 1 mM CellTrace™ Far Red (Invitrogen) to allow for fluorescent visualisation and downstream 3D modelling of the RBC. Stained RBCs at 6% haematocrit were treated with 1 mM cytochalasin D (ThermoFisher Scientific) or left untreated, then distributed as 75 mL aliquots in a 96 well plate. PfCERLI2^{HAGImS} ring-stages at 4-5% parasitaemia were treated with 2.5 mM GLCN or left untreated, incubated until young schizonts had formed at the point of chromatin segregation, Percoll-purified, then treated with E64 for 4 hours. Viable merozoites were purified as described previously with modifications²⁰: Percoll-purified schizonts were resuspended in 2 mL culture media excluding Albumax and passed through a 1.2 mm Minisart® Syringe Filter, and 75 mL of flowthrough was immediately added to wells containing CellTrace-stained, cytochalasin D treated or untreated RBCs, now at a final haematocrit of 3%. The plate was shaken at 300 rpm at 37 °C for 2 minutes to allow for a short invasion window, the cells were then fixed (final concentration 4% w/v paraformaldehyde (PFA, Sigma-Aldrich), 0.0075% v/v glutaraldehyde (Electron Microscopy Sciences), pH 7.4). Slides for confocal microscopy were prepared as described in a following section – stains included anti-AMA1 (mouse 1/500), anti-MSP1-19 (rabbit 1/1000) and DAPI. Slides were blinded shortly after sample preparation, and Z-stacks of 20 merozoites were taken per treatment group on an LSM800 (Zeiss) in Airyscan mode. Invasion was scored under three tiers: 1-attached, 2-partially invaded, 3-invaded, which could be determined by whether the merozoite appeared on the surface of, partway through the interface of, or enveloped inside the CellTrace fluorescently labelled RBC membrane respectively. The diameter of AMA1 of each merozoite was measured in Imaris (Bitplane). Samples were unblinded after statistical analyses were performed. Measurements for merozoites with an invasion score = 3 were excluded from the final data analysis as the tight junction closes behind the merozoite upon completion of invasion and becomes smaller. The full data set is provided in the supplementary material.”

Reference 20: Boyle, M. J. *et al.* Isolation of viable *Plasmodium falciparum* merozoites to define erythrocyte invasion events and advance vaccine and drug development. *Proceedings of the National Academy of Sciences* **107**, 14378-14383, doi:10.1073/pnas.1009198107 (2010).

Discussion:

Line 661: “Knockdown of PfCERLI2 led to reduced parasite growth and merozoite invasion. Morphology and rate of schizont rupture in PfCERLI2 knockdown parasites were indistinguishable compared to the controls, indicating merozoite development and egress was not impacted by knockdown of PfCERLI2. Instead, we observed increased numbers of both free merozoites in the culture medium and merozoites bound to the RBC surface in PfCERLI2 knockdown parasites. Based on this evidence, the inducible growth inhibition seen with PfCERLI2 knockdown is most likely to be a result of merozoite invasion being interrupted at or before formation of the irreversible tight junction with the RBC surface. This is further supported by a decrease in AMA1 diameter in merozoites bound to the RBC for PfCERLI2 knockdown parasites, suggesting that the invasion process had been halted at the early stages of host-cell entry and TJ formation. Indeed, a corresponding increase in RBC bound merozoites following PfCERLI2 knockdown suggests that rhoptry neck and micronemal protein contents were secreted for PfCERLI2 deficient merozoites, and the TJ formed for some of these, but these merozoites were then unable to proceed with invasion. The fact that we saw no reduction in secretion of the rhoptry neck proteins RON4 and RH4 following CERLI2 knockdown supports the idea that in many cases the glucosamine inducible ribozyme PfCERLI2 knockdown parasites can secrete enough protein to form the TJ, but these merozoites fail to invade due to a later-occurring defect.”

Reviewer # 1 minor comments:

1. Lines 56-60: A reference will be beneficial here for the non-expert reader.

Response: We agree with the reviewer that this section was lacking references. The section now reads:

Line 73: “...Each merozoite has two rhoptries, the largest of the invasion organelles, which are club-shaped and divided into a bulb and neck, with the neck positioned at the apical tip of the merozoite⁵. Rhoptry neck proteins are released early in the invasion process and mediate initial attachment to the RBC and formation of the tight junction⁶. Rhoptry bulb contents are secreted following junction formation and are typically involved in establishing the PV⁷.”

Reference 5: Hanssen, E. et al. Electron tomography of *Plasmodium falciparum* merozoites reveals core cellular events that underpin erythrocyte invasion. *Cellular Microbiology* **15**, 1457-1472, doi:doi:10.1111/cmi.12132 (2013).

Reference 6: Revealing the Sequence and Resulting Cellular Morphology of Receptor-Ligand Interactions during *Plasmodium falciparum* Invasion of Erythrocytes. *PLOS Pathogens* **11**, e1004670, doi:10.1371/journal.ppat.1004670 (2015).

Reference 7: Riglar, D. T. et al. Super-Resolution Dissection of Coordinated Events during Malaria Parasite Invasion of the Human Erythrocyte. *Cell Host & Microbe* **9**, 9-20, doi:10.1016/j.chom.2010.12.003 (2011).

2. Line 368: MSP1 does not have a transmembrane domain and text should be modified accordingly to “GPI-anchored”. Can the authors speculate on why is MSP1 detected in all fractions?

Response: This was an oversight on our behalf based on the current version of PlasmoDB. It annotates MSP1 as containing a transmembrane domain towards its C-terminus, although the presence of a GPI anchor is well established. We have changed the figure legend to reflect this, it now reads:

Line 425: “(b) PfCERLI2^{HAGImS} schizont lysates were subjected to differential lysis and solubilisation to determine the membrane solubility of PfCERLI2. Lysates were first hypotonically lysed, before being treated sequentially with sodium carbonate and triton-x-100. The supernatant was collected following each treatment, along with a final triton-x-100 soluble fraction. Each fraction was then probed with anti-HA (PfCERLI2), anti-GAPDH (cytosolic), anti-RH5 (peripheral) and anti-MSP1 (GPI-anchored) antibodies. Images representative of 3 independent experiments.”

To the best of our knowledge, no protein has been shown to be present exclusively in the Triton-x-100 soluble or insoluble fractions for *P. falciparum* membrane solubility assays. In the PfCERLI1 study we used EXP2 instead of MSP1, which produced very similar results (Liffner et al., 2020). In past solubility experiments in our lab we have also tried using integral membrane proteins such as NHE2 and obtain similar results where we get partial solubility in all fractions (unpublished). It is not clear what the cause of this is, perhaps a peculiarity of *Plasmodium* membranes.

Liffner, B. et al. PfCERLI1 is a conserved rhopty associated protein essential for Plasmodium falciparum merozoite invasion of erythrocytes. *Nature Communications* **11**, 1411, doi:10.1038/s41467-020-15127-w (2020).

- 3. Fig 4a. In most IFAs of early schizonts, the PfCERLI2 signal is only visible in few merozoites. Is this a technical issue, since the super resolution images show expression in all merozoites? Can the authors provide any better IFAs? On the AMA1 panel it seems that in the merged image AMA1 is shown as green instead of magenta. If the case, this should be corrected. On a similar note, AMA1 might not be the best micronemal marker as it is expressed quite late in the life cycle. Although not essential to this study, are there any other micronemal antibodies available that authors could use?**

Response: Response: The lack of PfCERLI2 signal in all merozoites in Figure 4a is potentially a product of both technical and biological factors. In Figure 4c, where all merozoites visible have PfCERLI2 staining, the images are 3D z-projections while in Figure 4a, where only some merozoites have visible PfCERLI2 staining, only a single z-slice was imaged. Therefore, it is possible that some of the PfCERLI2 foci are truly present but are not visible in the 2D plane imaged. The reason the colocalization imaging was performed only in 2D was due to the low expression of CERLI2 and the rapid photobleaching of its signal, as 2D imaging is much faster than 3D. In the text we have probably understated the difficulty in obtaining images even of this quality, which took over a year of optimisation and testing out different signal amplification methods before adopting the biotin-conjugated system. A section under “PfCERLI2 localises to the rhopty bulb” now reads:

Line 358: “...In our numerous attempts to localise PfCERLI2, we experienced that conventional antibody staining methods could not reliably visualise any foci directed to its HA-tag outside of background signal. After failing to obtain a consistent PfCERLI2 localisation, likely due to

the low peak expression level of Pfcerli2¹⁹, we investigated the cellular distribution of PfCERLI2 using a high affinity anti-HA biotin system to significantly bolster the signal²².”

Reference 19: Lopez-Barragan, M. J. *et al.* Directional gene expression and antisense transcripts in sexual and asexual stages of *Plasmodium falciparum*. *BMC Genomics* (2011).

Reference 22: Yao, Z. *et al.* Improved targeting of radiolabeled streptavidin in tumors pretargeted with biotinylated monoclonal antibodies through an avidin chase. *J Nucl Med* **36**, 837-841 (1995).

We suspect that many of the PfCERLI2 foci that are truly present are close to the limit of detection with the system and instruments we used and the low abundance of this protein may result in foci that are not optimally in the focal plane being just under the limit of detection.

The use of AMA1 as a micronemal marker was primarily because it doubles as a molecular clock for the age of the parasite. It was by attempting to colocalise AMA1 with PfCERLI2, we first identified that the younger schizonts with punctate AMA1 have visible PfCERLI2 foci (Figure 4a), while older schizonts with surface translocated AMA1 lack visible PfCERLI2 staining (Supplementary Figure 10). While we agree that the microneme to merozoite surface translocation and the later expression of AMA1 complicate its use as a micronemal marker, other micronemal markers, such as EBA175, do not allow us to determine the age of the schizont like AMA1 does.

We apologise for the miscolouring of AMA1 and PfCERLI2 in Figure 4a, this issue has been addressed in the updated manuscript.

4. Fig. 7i&j: Representative images as a supplement should be provided.

Response: Representative micrographs and fluorescence intensity plots for Figure 7i&j are included in Supplementary Figure 13. The Figure 7 legend has been updated to clarify this and now reads:

*“The fluorescence intensity from maximum-intensity projections of RAP1 (i) and RON4 (j) signals were then measured from the basal end of the nucleus. 32 merozoites for untreated and 31 merozoites for 2.5 mM GLCN treated were measured from a total of 12 schizont images across three biological replicates. Representative images corresponding to this data in Supplementary Figure 13. Error bars = SEM. *= $p < 0.05$, **= $p < 0.01$, ****= $p < 0.0001$ by unpaired t-test.”*

- 5. The loss of HA signal in late schizonts while still detectable in WB is intriguing. The authors speculate masking of the epitope. For this to occur it would suggest formation of a complex leading to the HA not being accessible. Do the authors have another possible explanation that they could speculate on? Relevant to this, in Figure 6a the HA western blot seems to suggest there is a doublet, but in other Western blots this doublet is not present. Is the doublet an artefact? Do the authors have any evidence for proteolytic processing of PfCERLI2?**

Response: We do not have any good alternative explanations other than conformational change in PfCERLI2 and/or complex formation masking the HA-epitope. A recent study,

however, showed that Protein Kinase A likely phosphorylates PfCERLI2 during schizogony (Patel et al., 2019) and so this phosphorylation event could either impart a conformational change, or promote complex formation, to mask the HA-epitope.

The doublet does not appear to be an artefact, as when wildtype Western blots were run in parallel to PfCERLI2^{HAGImS} parasites we never observed a background band in the same place. Despite this, it is not clear why we see the doublet, which can also be observed in Figures 2d&e. The pellet fraction of Figure 6a essentially represents the state of PfCERLI2 that is contained in free merozoites and the doublet may indeed indicate that PfCERLI2 is proteolytically processed at the N-terminus (any C-terminal processing would remove the HA-tag and prevent detection). The stronger doublet in Figure 6a could suggest that any potential proteolytic cleavage event would likely occur between merozoite egress and invasion. It does not appear likely, however, that this potential proteolytic cleavage event leads to a complex formation that masks the epitope from detection by IFA since the loss of PfCERLI2 signal occurs prior to schizont rupture and merozoite release

Patel, A. *et al.* Cyclic AMP signalling controls key components of malaria parasite host cell invasion machinery. *PLOS Biology* **17**(5): e3000264. <https://doi.org/10.1371/journal.pbio.3000264> (2019).

Lasonder, E *et al.* Extensive differential protein phosphorylation as intraerythrocytic *Plasmodium falciparum* schizonts develop into extracellular invasive merozoites. *Proteomics*. **15**(15):2716-29. doi: 10.1002/pmic.201400508 (2015).

- 6. The analysis on the repeats (lines 180-195) is not integral to the paper and should be compressed focusing on the ratio of mutations and the key points as no function can be attributed to them.**

Response: This section has been shortened from four sentences down to two, and now reads:

Line 204: “It is not clear what the function of this repeat is, or what drives changes in its number. However, we noticed that *P. falciparum* has a higher number of repeats than other *Laverania*, with organisms more closely related to *P. falciparum* also displaying higher repeat numbers than those more distantly related (Supplementary Figure 8; Supplementary Table 1).”

- 7. Supplementary Fig. 5. From the predicted modelling of the whole PfCERLI2 structure and the PH domain structure, I would be very cautious on describing the PH domain as degenerate and non-functioning. It seems as if it is completely absent. If the authors feel that it is a degenerate domain they should provide a superimposed image of both to show similarities in the fold. They should also highlight the N and C terminus in the models and the figure legend.**

Response: In Figure 5 and Figure 6 we provide predicted structures for the PfCERLI1 PH domain and the corresponding region of PfCERLI2. These clearly show the major structural differences predicted between these domains. They are so different that efforts to superimpose the two structures have failed to produce any meaningful comparison. So we agree with the reviewer that the PH domain is likely to have been effectively lost in PfCERLI2. However, there remains some minor similarity in sequence and structure between the PH domain of PfCERLI1 and the corresponding region of PfCERLI2. This suggests that, rather than

the PH domain being completely replaced in PfCERLI2 by the decapeptide repeat domain, the region once containing the PH domain remains but has undergone significant changes in amino acid sequence. In order to remove the potential ambiguity associated with the term degenerate, we have modified the following:

Results:

Line 173: *“By contrast, Plasmodium homologues of CERLI2 are predicted to have two beta-sheets in the region corresponding to the PH domain in CERLI2 of other species and PfCERLI1, but this region also has an expanded helix and disordered region between the beta-sheets (Supplementary Figure 6). Additionally, PfCERLI2 is predicted to contain a shorter C-terminal helix (Supplementary Figure 6). The expansion between the beta-sheets is predicted to disrupt the interaction between them that is required for canonical PH domain formation. Therefore, although elements of the PH domain are predicted to remain (beta-sheets) structural predictions do not annotate a PH domain structure in this region and it is questionable whether this domain structure is functional in PfCERLI2.”*

Discussion:

Line 736: *“The exception to this is the genus Plasmodium, where the region corresponding to the PH domain of PfCERLI1 in CERLI2 has undergone significant change in sequence and structure such that a PH domain is no longer predicted for, or likely functional in, this region of Plasmodium spp. CERLI2.”*

8. The inhibition of RAP1 processing is intriguing. Are both processing events inhibited? Can the authors check if Plasmepsin IX is still in the rhoptries or can they speculate on this part of the phenotype?

Response: It is thought that the first RAP1 processing event is performed by Plasmepsin IX, and the second by SUB1 (Nasamu et al., 2017, Silmon de Monerri et al., 2011, Favuzza et al., 2020). Broadly, the inhibition of RAP1 processing is similar between knockdown of PfCERLI2 and Plasmepsin IX, although the defect caused by Plasmepsin IX knockdown appears more severe (Nasamu et al., 2017). In both instances, knockdown causes a build-up of unprocessed RAP1 and a decrease in fully processed RAP1 with no noticeable change in the abundance of the intermediate product. This would imply that knockdown of either PfCERLI2 or Plasmepsin IX inhibit the processing of RAP1 by Plasmepsin IX, but any RAP1 that does get processed by Plasmepsin IX can subsequently be processed normally by SUB1. However, a direct comparison between RAP1 processing blots presented in this study and those in Nasamu et al., 2017, should be interpreted with caution as each use a different anti-RAP1 monoclonal (7H8 in this study and 2.29 in the Nasamu et al., 2017 study) and to the best of our knowledge the exact epitopes these antibodies recognise are not known.

Given that PfCERLI2 knockdown results in only a partial inhibition of RAP1 processing, compared to Plasmepsin IX knockdown which results in nearly complete inhibition of RAP1 processing, we suspect Plasmepsin IX is still present in the rhoptries. To the best of our knowledge, there is no suitable Plasmepsin IX specific antibody for us to check whether it still resides in the rhoptries at equal concentrations with PfCERLI2 knock-down. Moreover, since PfCERLI2 localises to the cytosolic face of the rhoptries, and Plasmepsin IX the rhoptry lumen (Nasamu et al., 2017), we suspect any inhibition of Plasmepsin IX processing caused by PfCERLI2 knockdown is indirect. While we have no evidence for how this indirect inhibition of

Plasmepsin IX might occur, it has been shown that phosphorylation of Plasmepsin IX is linked to Adenylate cyclase beta (AC β) and Protein Kinase A (PKA) (Patel et al., 2019). Like PfCERLI2, AC β has been shown to localise to the cytosolic face of the rhoptries and PKA is also involved in phosphorylation of PfCERLI2 (Patel et al., 2019). Thus, one possibility is that the aberrant rhoptry morphology caused by PfCERLI2 knockdown influences the ability of AC β and in turn PKA to lead to the phosphorylation of PMIX, inhibiting its processing of RAP1.

Nasamu, A. S. *et al.* Plasmepsins IX and X are essential and druggable mediators of malaria parasite egress and invasion. *Science* **358**, 518, doi:10.1126/science.aan1478 (2017).

Silmon de Monerri, N. C. *et al.* Global identification of multiple substrates for *Plasmodium falciparum* SUB1, an essential malarial processing protease. *Infect Immun* **79**, 1086-1097, doi:10.1128/IAI.00902-10 (2011).

Favuzza, P. *et al.* Dual Plasmepsin-Targeting Antimalarial Agents Disrupt Multiple Stages of the Malaria Parasite Life Cycle. *Cell Host & Microbe*, doi:10.1016/j.chom.2020.02.005 (2020).

Patel, A. *et al.* Cyclic AMP signalling controls key components of malaria parasite host cell invasion machinery. *PLOS Biology* **17**(5): e3000264. <https://doi.org/10.1371/journal.pbio.3000264> (2019).

Reviewer #2 comments:

- 1. I know that the authors are speaking generally, but the last half of the second paragraph of the Introduction could still use some references. Further, the use of a semicolon (;) in line 55 is unnecessary and would be better replaced with a comma (,).**

Response: This was also noted by Reviewer #1, three references have now been included in this section, which now reads:

Line 73: “Each merozoite has two rhoptries, the largest of the invasion organelles, which are club-shaped and divided into a bulb and neck, with the neck positioned at the apical tip of the merozoite⁵. Rhoptry neck proteins are released early in the invasion process and mediate initial attachment to the RBC and formation of the tight junction⁶. Rhoptry bulb contents are secreted following junction formation and are typically involved in establishing the PV⁷.”

Additionally, the semicolon has been replaced by a comma.

Reference 5: Hanssen, E. et al. Electron tomography of *Plasmodium falciparum* merozoites reveals core cellular events that underpin erythrocyte invasion. *Cellular Microbiology* **15**, 1457-1472, doi:doi:10.1111/cmi.12132 (2013).

Reference 6: Revealing the Sequence and Resulting Cellular Morphology of Receptor-Ligand Interactions during *Plasmodium falciparum* Invasion of Erythrocytes. *PLOS Pathogens* **11**, e1004670, doi:10.1371/journal.ppat.1004670 (2015).

Reference 7: Riglar, D. T. et al. Super-Resolution Dissection of Coordinated Events during Malaria Parasite Invasion of the Human Erythrocyte. *Cell Host & Microbe* **9**, 9-20, doi:10.1016/j.chom.2010.12.003 (2011).

2. Line 56 – “...are dual club-shaped” sounds incomplete.

Response: This sentence has been reworded as follows for clarity:

Line 73: “Each merozoite has two rhoptries, the largest of the invasion organelles, which are club-shaped and divided into a bulb and neck, with the neck positioned at the apical tip of the merozoite⁵.”

Reference 5: Hanssen, E. et al. Electron tomography of *Plasmodium falciparum* merozoites reveals core cellular events that underpin erythrocyte invasion. *Cellular Microbiology* **15**, 1457-1472, doi:doi:10.1111/cmi.12132 (2013).

3. Malaria phylogenetics isn't something I'm particularly well versed at, so I wasn't aware that Laverania was a subgenus of Plasmodium. I think it'd be helpful if there was a brief sentence about it for readers like myself.

Response: Until 2009 *Laverania* only comprised *P. falciparum* and *P. reichenowi*, so it is likely that many readers would also be unfamiliar with the classification. To reflect this, we have included a brief description of *Laverania* in the sentence where they are first mentioned, which reads:

Line 107: “*PfCERLI2* is highly conserved amongst *Laverania*, a subgenus that comprises *P. falciparum* and its closest ape-infecting relatives (>90% amino acid identity with *Laverania* homologues), and shares approximately 65% amino acid identity amongst more distantly related *Plasmodium* spp. (Figure 1b) (Supplementary Figure 2).”

4. Line 118 “...sequence identities werelow...”

Response: This error has been corrected.

5. Is the growth of PfCERLI2HAGImS with glucosamine delayed compared to no glucosamine? Also, do the treated parasites morphologically appear normal throughout the life cycle prior to invasion?

Response: It does not appear that either GLCN treatment itself or PfCERLI2 knockdown results in significant delays to growth. In Figure 3c it can be seen that 2.5 mM GLCN treatment of PfCERLI2^{HAGImS} parasites does not alter the timing of schizont rupture, and if GLCN or knockdown delayed growth we would expect less rupture in the + GLCN treatment. In Figure 2f it can be seen that, while there is a growth defect notable for wildtype parasites treated with 5 mM GLCN, at 2.5 mM GLCN, the concentration used throughout this study, any delay to growth is negligible. We have also included representative Giemsa images of PfCERLI2^{HAGImS} parasites at rings and trophozoites +/- 2.5 mM GLCN in Supplementary Figure 9 a&b where we observed no obvious morphological differences in response to treatment across a 24 hour treatment window.

A section under “PfCERLI2 knockdown inhibits merozoite invasion” now reads:

Line 306: “Knockdown of PfCERLI2 did not result in a change in the morphological development of trophozoites (Supplementary Figure 9 a&b), nor in the number of fully formed merozoites per schizont (Figure 3b).”

6. The authors state that expression of PfCERLI2 was reduced by ~72%, based upon the western blot. How exactly was this number derived? I’m assuming that bands were normalized to the loading control (ERC in this case) and quantified, but I didn’t see an indication of how many biological replicates. Also, do the authors know if all parasites show the same level of reduction? If a treated parasite still has a little PfCERLI2, is this enough to help it proceed through invasion?

Response: This was an oversight on our behalf, we have now included a graph (below the original Figure 2e) that quantifies the knockdown of PfCERLI2 and displays that X replicates were performed to obtain the 72% number quoted in the text. Exactly as suggested, this quantification was based on normalisation against ERC. The legend now reads:

Line 255: “...(e) Western blot of PfCERLI2^{HAGImS} schizont lysates either GLCN treated (+) or untreated (-) and probed with anti-HA (PfCERLI2) normalised to anti-ERC (loading control) antibodies. A graph quantifying the relative level of expression between treatments. N=3 biological replicates.”

In regard to the dynamics of knockdown, this question is difficult to answer but it seems your assumption is likely. In a similar study of PfCERLI1, also using the *GImS* ribozyme system, we showed by microscopy that 97% of PfCERLI1 knockdown parasites had a PfCERLI1 signal intensity below the mean of untreated parasites (Liffner et al., 2020). Notably, PfCERLI1 knockdown led to a similar level of invasion inhibition as we see for PfCERLI2 knockdown. Considering that expression is reduced in nearly all parasites, we suspect there may be a threshold of PfCERLI2 expression where above that threshold invasion can still proceed normally, and below it invasion is likely to fail.

Liffner, B. *et al.* PfCERLI1 is a conserved rhopty associated protein essential for Plasmodium falciparum merozoite invasion of erythrocytes. *Nature Communications* **11**, 1411, doi:10.1038/s41467-020-15127-w (2020).

7. I was confused by the addition of the PfCERLI2 growth curve (first pictured in Fig 2F) to Fig 3a. I don’t think this is necessary and ideally, I would have preferred to see if WT treated with Glcn showed no differences in invasion.

Response: The growth data was initially included in Figure 3a to have a side-by-side comparison of growth and invasion inhibition, but to save confusion it has now been omitted from Figure 3a. The Figure 3 legend has now been modified to read as follows:

“Figure 3: PfCERLI2 knockdown inhibits merozoite invasion.

Line 277: a) GFP-expressing PfCERLI2^{HAGImS/GFP} parasites were treated with a range of concentrations of GLCN from early trophozoite stages until early rings the following cycle (~24 hours), with invasion assessed by flow cytometry as the number of newly invaded rings expressed as a percentage of an untreated media control. n=4, error bars = SEM. X-axis presented as log 2 scale for viewing purposes.”

In the invasion assay we used GFP-expressing parasites to identify newly-invaded ring stage parasites by flow cytometry. We do not have a GFP-expressing wild-type 3D7 parasite line to compare with as the GFP plasmid used does not maintain as an episome in 3D7. However, we have shown across this and the PfCERLI1 study that 2.5 mM GLCN treatment has a negligible influence on parasite growth which includes one round of merozoite invasion.

- 8. The authors note that the invasion assay had a shorter Glcn treatment, compared to the growth assay (24h vs 72h) and that this may be a reason for reduction being lower than growth inhibition (30% invasion reduction vs ~50% growth inhibition). The western blot (Fig2e) was accomplished by treating ring stage to schizonts (~48h) (line 209), from which expression was said to be reduced by 72%. Could the authors explain why for the invasion assay they did not do a similar treatment (ring-stage to ring stage of the next cycle? What are the expression levels of PfCERLI2 in parasites treated with Glcn from trophozoite to ring-stage (24h treatment)? Further, were successful invasions after Glcn treatment able to progress normally through the next life cycle or does a % not grow properly after invasion? Sidenote: I feel that this is a disadvantage in using the ribozyme system for exploring protein functions and think that the use of a conditional knock out system (e.g., DiCre) would have been better in getting a clearer idea of the effects of deficiency.**

Response: The reason for the shorter treatment time when testing the effect of PfCERLI2 knock-down on merozoite invasion directly compared to the Western blot for expression was to target PfCERLI2 knockdown as close to when the protein is actually expressed and the parasite's invasion window as possible, minimising any potential off-target effects at invasion. Assuming the knockdown of PfCERLI2 takes some time to take effect following GLCN treatment, the 24 hr troph to ring-stage treatment gave us the best ability to limit the window of knockdown to merozoite invasion while also treating with GLCN for long enough to see a defect. Nevertheless, we agree with the reviewer and have examined the level of GLCN-inducible knockdown over 24 hours from trophozoites to schizonts. A western blot is now included in Supplementary Figure 9b and after quantification of three biological replicates we found that treatment of PfCERLI2^{HAGImS} trophozoites with 2.5 mM GLCN and harvesting at schizonts resulted in a 65% knockdown in HA-tagged PfCERLI2 protein which is a marginal decrease when compared to 72% knockdown over 48 hours. This difference in knock-down levels with the shorter treatment time may well contribute to the lower invasion inhibition we see in the merozoite invasion assays. However, it is also reassuring that we were able to achieve a similar knockdown in PfCERLI2 protein levels with the shorter treatment time and that the invasion defect reflects the overall loss in growth with longer treatment times.

We agree that the use of the DiCre inducible knockout system could overcome some of the intricacies of these kinds of experiments, but the system was unfortunately not widely available at the time this study started and the glucosamine inducible ribozyme protein knock-down system has proven to be a robust method to investigate protein function across a number of original protein characterisation studies.

9. Please clarify the treatment time for the merozoite quantity experiment.

Response: These parasites were treated with GLCN from ring-stages to schizont stages in the same cycle. This information was previously included in the methods section but has now been included into the legend of Figure 3 for clarity. The Figure 3 legend now reads:

Line 281: “...(b) *PfCERLI2^{HAGImS}* parasites were either treated with GLCN from ring-stage to schizont-stage or left untreated. Early schizonts were treated with the egress inhibitor E64 for ~5 hours. Following treatment, cultures were smeared and the number of merozoites per schizont was determined by blinded microscopy analysis of Giemsa-stained blood smears. Each data point represents the mean number of merozoites per schizont from 20 schizonts. $n=3$, error bars = SEM, $ns=p>0.05$.”

10. Invasion is affected as evidenced by the decrease in number of successful invasion events, however it is unclear to me at which point invasion is affected. Have the authors considered live imaging to measure the duration of pre-invasion between treated/untreated parasites? Do the merozoites bind initially to RBCs and then fall off? Or do the parasites remain bound to the RBC membrane?

Response: Please see our response to Reviewer 1, comment #4 regarding our updated approach to understanding the temporal placement of PfCERLI2's function.

11. In Fig 5a (proteinase protection assay), the authors use ERC as a control to show that organelle contents are unaffected. Have the authors also tested a rhoptry protein (e.g., RAP1) to show that the rhoptry contents itself are not affected by proteinase treatment? I would think that this be the more appropriate control given that you're asserting that PfCERLI2 is outside, not inside, the rhoptries. Additionally, there appears to be less ERC after proteinase treatment, compared to no treatment. Could the authors comment on this?

Response: We have taken the reviewers suggestion and included the rhoptry luminal protein RAP1 as a proteinase K protected control in Figure 5a. In this blot RAP1 is still present in proteinase K-treated samples, however much like ERC we observe some slight degradation of protein. We note that minor levels of cleavage for organelle localised control proteins is common in proteinase-K protection experiments (Liffner *et al.*, 2020; Grüning *et al.*, 2012). We hypothesise that one reason for this is digitonin could be capable of weakly permeabilising organelle membranes at the concentration used and/or with the multiple processing steps required for this assay, as it has been previously shown that increasing concentrations of digitonin can permabilise rhoptries in *Toxoplasma* (Suarez *et al.*, 2019). Or proteinase K itself may be able to damage membranes in a way that does not require permeabilisation to access the lumen. Another possibility is that protein being transported from the Golgi to the rhoptry lumen may be susceptible to solubilisation and cleavage in the case of RAP1. Again, degradation for rhoptry organellar proteins is often seen in these assays.

Liffner, B. *et al.* PfCERLI1 is a conserved rhoptry associated protein essential for Plasmodium falciparum merozoite invasion of erythrocytes. *Nature Communications* **11**, 1411, doi:10.1038/s41467-020-15127-w (2020).

Grüning, C. *et al.* Uncovering Common Principles in Protein Export of Malaria Parasites. *Cell Host & Microbe* **12**(5): 717-729 (2012).

Suarez, C. *et al.* A lipid-binding protein mediates rhoptry discharge and invasion in *Plasmodium falciparum* and *Toxoplasma gondii* parasites. *Nature Communications* **10**, 4041, doi:10.1038/s41467-019-11979-z (2019).

12. The authors have used flow cytometry in a number of these studies; however, I have not seen any representative plots to show their gating strategies.

Response: We have now included representative flow plots in Supplementary Figure 17 to show our gating strategies for RBCs, merozoites, late-stage parasites, and GFP-fluorescent ring stages.

The methods under “Assessment of *in vitro* blood stage growth and invasion” now reads:

Line 875: “Completed assays were then stained in 10 mg/mL ethidium bromide (Bio-Rad) in PBS and the final parasitaemia was assessed using the BD Accuri™ C6 Plus flow cytometer (PE-H, FSC-H filters for growth, PE-H, FITC-H for invasion to visualise GFP-fluorescent ring stages). All flow cytometry files were analysed using FlowJo Version 10 (Tree Star). Representative flow plots are included in Supplementary Figure 17 to show our gating strategies.”

REVIEWERS' COMMENTS:

Reviewer #1 (Remarks to the Author):

The revised manuscript includes new data regarding potential interactors of PfCERLI1 and 2 and super-resolution microscopy to clarify the invasion phenotype observed in PfCERLI2 knockdown parasites. Overall, the authors have very comprehensively addressed all my comments and I recommend that this manuscript is accepted for publication.

Reviewer #2 (Remarks to the Author):

The authors have addressed all concerns and I have no further comments.